# Escaping Saddle-Point Faster under Interpolation-like Conditions

**Abhiskek Roy**
Department of Statistics
University of California, Davis
abroy@ucdavis.edu

**Krishnakumar Balasubramanian**
Department of Statistics
University of California, Davis
kbala@ucdavis.edu

**Saeed Ghadimi**
Department of Management Sciences
University of Waterloo
sghadimi@uwaterloo.ca

**Prasant Mohapatra**
Department of Computer Science
University of California, Davis
pmohapatra@ucdavis.edu

## Abstract

In this paper, we show that under over-parametrization several standard stochastic optimization algorithms escape saddle-points and converge to local-minimizers much faster. One of the fundamental aspects of over-parametrized models is that they are capable of interpolating the training data. We show that, under interpolation-like assumptions satisfied by the stochastic gradients in an over-parametrization setting, the first-order oracle complexity of Perturbed Stochastic Gradient Descent (PSGD) algorithm to reach an $\epsilon$-local-minimizer, matches the corresponding deterministic rate of $\tilde{\mathcal{O}}(1/\epsilon^2)$. We next analyze Stochastic Cubic-Regularized Newton (SCRN) algorithm under interpolation-like conditions, and show that the oracle complexity to reach an $\epsilon$-local-minimizer under interpolation-like conditions, is $\tilde{\mathcal{O}}(1/\epsilon^{2.5})$. While this obtained complexity is better than the corresponding complexity of either PSGD, or SCRN without interpolation-like assumptions, it does not match the rate of $\tilde{\mathcal{O}}(1/\epsilon^{1.5})$ corresponding to deterministic Cubic-Regularized Newton method. It seems further Hessian-based interpolation-like assumptions are necessary to bridge this gap. We also discuss the corresponding improved complexities in the zeroth-order settings.

## 1 Introduction

Over-parametrized models, for which the training stage involves solving nonconvex optimization problems, are common in modern machine learning. A canonical example of such a model is deep neural networks. Such over-parametrized models have several interesting statistical and computational properties. On the statistical side, such over-parametrized models are highly expressive and are capable of nearly perfectly interpolating the training data. Furthermore, despite the highly nonconvex training landscape, most local minimizers have good generalization properties under regularity conditions; see for example [39, 27, 22, 21] for empirical and theoretical details. We emphasize here that over-parametrization plays an important role for both phenomenon to occur. Furthermore, it is to be noted that not all critical points exhibit nice generalization properties. Hence, from a computational perspective, designing algorithms that do not get trapped in saddle-points, and converge to local minimizers during the training process, becomes extremely important [12].

Indeed, recently there has been extensive research in the machine learning and optimization communities on designing algorithms that escape saddle-points and converge to local minimizers. The

authors of [30] proved the folklore result that in the deterministic setting for sufficiently regular functions, vanilla gradient descent algorithms converges almost surely to local minimizers, even when initialized randomly; see also [29]. However, [30, 29] only provide asymptotic results, that have limited consequence for practice. Understandably, it has been shown by the authors of [15], that gradient descent might take exponential-time to escape saddle points in several cases. In this context, injecting artificial noise in each step of the gradient descent algorithm has been empirically observed to help escape saddle points. Several works, for example, [24, 26], showed that such *perturbed* gradient descent algorithms escape saddles faster in a non-asymptotic sense. Such algorithms are routinely used in training highly over-parametrized deep neural network and other over-parameterized nonconvex machine learning models. However, existing theoretical analysis of such algorithms fail to take advantage of the interpolation-like properties enjoyed by over-parametrized machine learning models. Hence, such theoretical results are conservative. Specifically, there is a gap between the assumptions used in the theoretical analysis of algorithms that escape saddle-points and the assumptions commonly satisfied by over-parametrized models which are trained by those algorithms.

In this work, we consider nonconvex stochastic optimization problems of the following form:

$$\underset{x \in \mathbb{R}^d}{\operatorname{argmin}} \left\{ f(x) := \mathbf{E}_\xi[F(x, \xi)] \right\}. \tag{1}$$

where $f : \mathbb{R}^d \rightarrow \mathbb{R}$ is nonconvex function satisfying certain regularity properties described next, and $\xi$ is a random variable characterizing the stochasticity in the problem. We assume that the function $f$ has a lower bound $f^*$ throughout this work. We analyze two standard algorithms that escape saddle-points, namely the perturbed stochastic gradient descent (PSGD) and stochastic cubic-regularized Newton's method (SCRN) for problems of the form in (1). We show that under interpolation-like assumptions (see Section 2 for exact definitions) on the stochastic gradient, it could be proved that both PSGD and SCRN escape saddle-points and converge to local minimizers much faster. In particular, we show that in order for PSGD algorithm to escape saddle-points and find an $\epsilon$-local-minimizer, the number of calls to the stochastic first-order oracle is of the order $\tilde{\mathcal{O}}(1/\epsilon^2)$[1] which matches number of calls when the objective being optimized is a deterministic objective (for which exact gradient could be obtained in each step of the algorithm)[2]. As a point of comparison, [19, 25] showed that without the interpolation-like conditions that we make, PSGD requires $\tilde{\mathcal{O}}(1/\epsilon^4)$ calls to the stochastic gradient oracle. Furthermore, [17] analyzed a version of PSGD with averaging and improved the oracle complexity to $\tilde{\mathcal{O}}(1/\epsilon^{3.5})$. It is also worth noting that, with a mean-square Lipschitz gradient assumption on the objective function being optimized, and using complicated variance reduction techniques, the authors of [16] showed that it is possible for a double-loop version of PSGD to converge to $\epsilon$-local minimizers with $\tilde{\mathcal{O}}(1/\epsilon^3)$ number of calls to the stochastic first-order oracle. However, recent empirical investigations seem to suggest that variance reduction techniques are inefficient for the nonconvex deep learning problems [13, 43]. Our results, on the other hand exploit the naturally available structure present in over-parametrized models and obtains the best-known oracle complexity for escaping saddle-points using only the vanilla versions of PSGD algorithm (which is oftentimes the version of PSGD used in practice). We also analyze the corresponding Zeroth-Order version of the PSGD algorithm. In this setting, we are able to observe only potentially noisy evaluations of the function being optimized. In this setting, we show that PSGD algorithm requires $\tilde{\mathcal{O}}(d^{1.5}/\epsilon^{4.5})$ calls to the stochastic zeroth-order oracle. In this context, we are not aware of a result to compare with. The recent works of [4, 18] provided results for bounded functions in the zeroth-order deterministic setting, where one obtains exact function values; such a setting though is highly unrealistic in practice.

Next, we consider the question of whether using second-order methods helps reduce the number of calls. Indeed, in the deterministic setting, it is well-known that second-order information helps escape saddle point at a much faster rate. For example, [37] proposed that Cubic-regularized Newton's method and showed that the method requires only $\tilde{\mathcal{O}}(1/\epsilon^{1.5})$ calls to the gradient and Hessian oracle; see also [8, 11] for related results. Correspondingly, in the stochastic setting [47] showed that SCRN method requires $\tilde{\mathcal{O}}(1/\epsilon^{3.5})$ calls, which is better than that of PSGD (without further assumptions). In this work, we show that under interpolation-like assumptions on (only) the stochastic gradient, SCRN method requires only $\tilde{\mathcal{O}}(1/\epsilon^{2.5})$ calls. In contrast to the PSGD setting, SCRN requires more calls than its corresponding deterministic counterpart. However, it should be noted that the complexity of SCRN

| Algorithm | With SGC (**This paper**) | | Without SGC | | Deterministic |
|---|---|---|---|---|---|
| | ZO | HO | ZO | HO | HO |
| Perturbed GD | $\tilde{\mathcal{O}}\left(d^{1.5}\epsilon^{-4.5}\right)$ Theorem 3.1 | $\tilde{\mathcal{O}}\left(\epsilon^{-2}\right)$ Theorem 3.1 | $\tilde{\mathcal{O}}\left(d^{1.5}\epsilon^{-5.5}\right)$ Theorem 3.2 | $\tilde{\mathcal{O}}\left(\epsilon^{-4}\right)$ Theorem 17 [25] | $\tilde{\mathcal{O}}\left(\epsilon^{-2}\right)$ Theorem 3 [24] |
| Cubic Newton | $\tilde{\mathcal{O}}\left(d^{4}\epsilon^{-2.5}\right)$ Theorem 4.1 | $\tilde{\mathcal{O}}\left(\epsilon^{-2.5}\right)$ Theorem 4.1 | $\tilde{\mathcal{O}}\left(d^{4}\epsilon^{-2.5}\right)+\mathcal{O}\left(d\epsilon^{-3.5}\right)$ Theorem 4.1 [5] | $\tilde{\mathcal{O}}\left(\epsilon^{-3.5}\right)$ Theorem 1 [47] | $\mathcal{O}\left(\epsilon^{-1.5}\right)$ Theorem 3 [37] |

Table 1: Oracle complexities of perturbed stochastic gradient descent (PSGD) and stochastic cubic-regularized Newton's method (SCRN). ZO corresponds to number of calls to zeroth-order oracle and HO corresponds to number of calls to first or second-order oracles. The result for PSGD and SCRN are given respectively in high-probability and in expectation. The results in the deterministic case corresponds to projected gradient descent and cubic-Regularized Newton's method (without stochastic gradients).

is still better than that of the PSGD, with or without interpolation-like assumptions. We belive that without further interpolation-like assumptions also on the stochastic Hessians, the oracle complexity of SCRN cannot be improved, in particular to match the deterministic rate of $\tilde{\mathcal{O}}(1/\epsilon^{1.5})$ (see also Remark 6). We also provide similar improved results for a zeroth-order version of SCRN method, thereby improving upon the results of [5]. All of our results, along with comparison to existing results in the literature and the corresponding assumption required, are summarized in Table 1. We conclude this section with a other related works.

**More Related Works.** In the interpolation regime, [33] recently showed that mini-batch stochastic gradient descent (SGD) algorithm enjoys exponential rates of convergence for unconstrained strongly-convex optimization problems; see also [45, 36] for related earlier work. For the non-convex setting, [6] analyze SGD for non-convex functions satisfying the Polyak-Lojasiewicz (PL) inequality [41] under the interpolation condition and show that SGD can achieve a linear convergence rate. Recently, [48] introduced a more practical form of interpolation condition, and prove that the constant step-size SGD can obtain the optimal convergence rate for strongly-convex and smooth convex functions. They also show the first results in the non-convex setting that the constant step-size SGD can obtain the deterministic rate in the interpolation regime for converging to first-order stationary solution. Subsequently, [34] investigate the regularized subsampled Newton method (R-SSN) and the stochastic BFGS algorithm under the interpolation-like conditions. We emphasize that all the above works consider the case of convex objective function predominantly; the only exception is [48] that consider the nonconvex case but only study convergence to first-order stationary solution. There has been several works on obtaining oracle complexity of escaping saddle-points in the finite-sum setting; we refer the interested reader to [1, 51, 50, 49] and references therein for such results. We emphasize that a majority of the above works are based on complicated variance reduction techniques that increase the implementation complexity of such methods and make them less appealing in practice. There exist only few works on escaping saddle-points for constrained optimization problems; see [32, 31, 40, 35] for more details.

We also briefly discuss the consequences of our results to deep neural network training and related works. Roughly speaking, there are now two potential explanations for the success of optimization methods for training deep neural networks [46]. The first explanation is based on landscape analysis. This involves two steps: Showing the optimization landscape has favorable geometry [28] (i.e., all local minima are (approximate) global minima under suitable regularity conditions), and hence constructing optimization algorithms that can efficiently escape saddle-points. The second explanation is based on the NTK viewpoint; see, for example [23, 9, 10, 3, 14, 52], for a partial overview. However, a majority of the results based on NTK viewpoint are for polynomially (in depth and sample-size) large-width networks (indeed, [3] mention that their polynomial degrees are impractical). Our results in this paper are geared towards the former program.

# 2 Preliminaries

We now present the assumptions and definitions used throughout the paper. Section-specific additional details are in the respective sections. In this paper we use $\|\cdot\|$, and $\|\cdot\|_*$ to denote a norm and the corresponding dual norm on $\mathbb{R}^d$. We now describe some regularity conditions made on the objective function in (1) assumptions in this work.

**Assumption 2.1 (Lipschitz Function)** *The function $F$ is $L$-Lipschitz, almost surely for any $\xi$, i.e., $|F(x,\xi) - F(y,\xi)| \leq L\|x-y\|$. Here we assume $\|\cdot\| = \|\cdot\|_2$, unless specified explicitly.*

**Assumption 2.2 (Lipschitz Gradient)** *The function $F$ has Lipschitz continuous gradient, almost surely for any $\xi$, i.e., $\|\nabla F(x,\xi) - \nabla F(y,\xi)\| \leq L_G\|x-y\|_*$, where $\|\cdot\|_*$ denotes the dual norm of $\|\cdot\|$. This also implies $|F(y,\xi) - F(x,\xi) - \nabla F(x,\xi)^\top(y-x)| \leq \frac{L_G}{2}\|y-x\|^2$.*

**Assumption 2.3 (Lipschitz Hessian)** *The function $F$ has Lipschitz continuous Hessian, almost surely for any $\xi$, i.e., $\left\|\nabla^2 F(x,\xi) - \nabla^2 F(y,\xi)\right\| \leq L_H\|x-y\|$.*

Note that if Assumptions 2.1–2.3 are true for $F$, then they also hold for $f(\cdot) = \mathbf{E}[F(\cdot,\xi)]$; but the other way around is not true. For our higher-order results, we make the above assumptions only on $f(\cdot)$, which is a weaker assumption. In the interpolation regime, the stochastic gradients become small when the true gradient is small. The following condition, known as Strong Growth Condition (SGC) [48], captures how fast the stochastic gradient goes to $0$ with respect to the true gradient.

**Assumption 2.4 (SGC [48])** *For any point $x \in \mathbb{R}^d$, the stochastic gradient satisfies $\mathbf{E}_\xi\|\nabla F(x,\xi)\|^2 \leq \rho\|\nabla f(x)\|^2$, for $\rho > 1$ (as $\rho = 1$, corresponds to the deterministic setting).*

SGC controls the variance of the obtained stochastic gradient in the above mentioned way. Note in particular that in the case when $\|\nabla f(x)\|^2 = 0$, under SGC, we have almost surely $\|\nabla F(x,\xi)\|^2 = 0$. This means that when the point $x$ is a stationary point of the function $f$, then it is also a stationary point of the function $F$ almost surely. In the context of deep neural networks, the function $F$ corresponds to the risk based on training sample $\xi$ and the function $f$ corresponds to the risk. Hence, the strong growth condition states that that deep neural network is capable of interpolating the training data almost surely. Such a phenomenon is observed in practice with deep neural networks, which provides a strong motivation for using this assumption for analyzing the performance of PSGD and SCRN for escaping saddle-points.

In this work, we study the algorithms under two oracles settings: Stochastic zeroth-order oracle, where one obtains noisy unbiased function evaluations, and the stochastic higher-order oracle, where one obtains noisy unbiased estimators of the gradients, and hessians. We now define them formally.

**Assumption 2.5 (Zeroth-order oracle)** *For any $x \in \mathbb{R}^d$, the zeroth order oracle outputs an estimator $F(x,\xi)$ of $f(x)$ such that $\mathbf{E}[F(x,\xi)] = f(x)$, $\mathbf{E}[\nabla F(x,\xi)] = \nabla f(x)$, $\mathbf{E}[\nabla^2 F(x,\xi)] = \nabla^2 f(x)$, and $\mathbf{E}[\|\nabla^2 F(x,\xi) - \nabla^2 f(x)\|_F^4] \leq \sigma_2^4$, where $\|\cdot\|_F$ is the Frobenius norm.*

**Assumption 2.6 (Higher-order oracles)** *For any $x \in \mathbb{R}^d$, (i) the first-order oracle outputs an estimate $\nabla F(x,\xi)$ of $\nabla f(x)$ such that $\mathbf{E}[\nabla F(x,\xi)] = \nabla f(x)$ and (ii) the second-order oracle, in addition outputs an estimate $\nabla^2 F(x,\xi)$ of $\nabla^2 f(x)$ such that, $\mathbf{E}[\nabla^2 F(x,\xi)] = \nabla^2 f(x)$, and $\mathbf{E}[\|\nabla^2 F(x,\xi) - \nabla^2 f(x)\|_F^4] \leq \sigma_2^4$.*

Such assumptions on the zeroth-order and higher-order oracles are standard in the literature; see for example [20, 33? ]. Our goal in this paper is to reach an approximate local minimizer (also called as a second-order stationary point) of a non-convex function, which is defined as follows:

**Definition 2.1 ($\epsilon$-Local Minimizer)** *Let Assumption 2.3 hold for a function $f$. Then a point $\bar{x}$ is called a $\epsilon$-second-order stationary point if,*

$$\max\left(\sqrt{\|\nabla f(\bar{x})\|}, -\frac{\lambda_{min}(\nabla^2 f(\bar{x}))}{L_H}\right) \leq \sqrt{\epsilon} \tag{2}$$

*where $\lambda_{min}(\nabla^2 f(\bar{x}))$ is the minimum eigenvalue of $\nabla^2 f(\bar{x})$.*

---

**Algorithm 1** Perturbed Stochastic Gradient Descent Algorithm

---

**Input:** $x_0 \in \mathbb{R}^d$, $\eta$, $r$.
**for** $t = 0$ to $T$ **do**
**Set** $g_t = \frac{1}{n_1} \sum_{i=1}^{n_1} g_{t,i}$ **where**

$$g_{t,i} = \nabla F(x_t, \xi_{t,i}) \qquad \text{(First-order)}$$

$$g_{t,i} = \frac{F(x_t + \nu u_{t,i}, \xi_{t,i}) - F(x_t, \xi_{t,i})}{\nu} u_i \qquad \text{(Zeroth-order)}$$

**and** $u_{t,i} \sim \mathcal{N}(\mathbf{0}, \boldsymbol{I}_d)$ $\forall t = 1, 2, \cdots, T$, $i = 1, 2, \cdots, n_1$
**Sample** $\theta_t \in \mathcal{N}(\mathbf{0}, r^2 \boldsymbol{I}_d)$
**Update** $x_{t+1} = x_t - \eta(g_t + \theta_t)$
**end for**

---

Note that for stochastic optimization problems, the quantity on the left hand side of (2), is a random variable. In this paper we prove a high-probability bound, and an expectation bound for the above quantity for PSGD, and SCRN respectively.

For a point $x_t$, we will use $\nabla_t$, $\nabla_t^2$, $h_t$, and $\lambda_{1,t}$ to denote $\nabla_t$, $\nabla^2 f(x_t)$, $(x_{t+1} - x_t)$, and $\lambda_{min}(\nabla^2 f(x_t))$ respectively. The zeroth-order minibatch gradient [38], and Hessian estimator [? ] $g_t$, and $H_t$ are defined as:

$$g_t = \frac{1}{n_1} \sum_{i=1}^{n_1} \frac{F(x_t + \nu u_{t,i}, \xi_{t,i}) - F(x_t, \xi_{t,i})}{\nu} u_i, \qquad H_t = \frac{1}{n_2} \sum_{i=1}^{n_2} \mathfrak{H}_{t,i} \left( u_{t,i} u_{t,i}^\top - I \right), \quad (3)$$

where $\mathfrak{H}_{t,i} = \frac{F(x_t + \nu u_{t,i}, \xi_{t,i}) + F(x_t - \nu u_{t,i}, \xi_{t,i}) - 2F(x_t, \xi_{t,i})}{2\nu^2}$, and $u_{t,i} \sim \mathcal{N}(\mathbf{0}, \boldsymbol{I}_d)$ $\forall t = 1, 2, \cdots, T$, $i = 1, 2, \cdots, n_1$. We will use $\zeta_t = g_t - \nabla_t = \frac{1}{n_1} \sum_{i=1}^{n_1} g_{t,i} - \nabla_t$, and $\tilde{\zeta}_t = \zeta_t + \theta_t$. In the following lemma we show that under SGC, the variance of $\nabla F(x_t, \xi)$ is of the order of the gradient norm squared.

**Lemma 2.1** *Let Assumption 2.4 hold for a function $f$. Then, for both zeroth-order, and first-order oracle, we have,*

$$\mathbf{E} \left[ \left\| \frac{1}{n_1} \sum_{i=1}^{n_1} \nabla F(x_t, \xi_i) - \nabla_t \right\|^2 \right] \leq \frac{\rho - 1}{n_1} \|\nabla_t\|^2. \qquad (4)$$

**Proof** Let $g_t = \frac{1}{n_1} \sum_{i=1}^{n_1} \nabla F(x_t, \xi_i)$. Then we have

$$\mathbf{E} \left[ \|g_t - \nabla_t\|^2 \right] = \mathbf{E} \left[ \|g_t\|^2 + \|\nabla_t\|^2 - 2g_t^\top \nabla_t \right] = \frac{1}{n_1^2} \mathbf{E} \left[ \left\| \sum_{i=1}^{n_1} \nabla F(x_t, \xi_i) \right\|^2 \right] - \|\nabla_t\|^2$$

$$\leq \frac{1}{n_1^2} \left( \rho n_1 \|\nabla_t\|^2 + n_1(n_1 - 1) \|\nabla_t\|^2 \right) - \|\nabla_t\|^2 = \frac{\rho - 1}{n_1} \|\nabla_t\|^2,$$

which completes the proof. ∎

**Remark 1** *The above simple results actually turns out to have far-reaching consequences for obtaining improved complexity bounds for both PSGD and SCRN algorithms. It implies that when the true gradient is small, the variance of the stochastic gradient is also small. Typically, in the analysis of PSGD and SCRN, it is assumed that the stochastic gradients are assumed to have a constant variance. But for over-parametrized models, we will use Lemma 2.1 to prove deterministic rate for PSGD and improved rates for SCRN.*

## 3 Perturbed Stochastic Gradient Descent

In this section we show that under SGC, PSGD attain deterministic rate in the first-order setting and obtains much better rate than previously known rates in the zeroth-order setting. An intuitive

explanation of this phenomenon is as follows: in the general stochastic setting, at time $t$ where $\|\nabla_t\| \geq \epsilon$, PSGD does not descend as much as in the deterministic setting due to noisy gradient. So it takes more iterations to average out the noise. While escaping a saddle point, due to noisy gradient, the iterates follow the direction of the most negative curvature with more difficulty leading to higher complexity. Under SGC, when $\|\nabla_t\| \geq \epsilon$, the noise variance is of the order of $\|\nabla_t\|^2$ as shown in Lemma 2.1. So the algorithm still manages to descent. While escaping a saddle point under SGC, as $\|\nabla_t\| \leq \epsilon$, and the gradient noise is also small leading to deterministic rates.

The outline of the proof of the bounds for PSGD in the first-order setting is similar to [25] except that we analyze PSGD under interpolation regime. At a high level the proof has two stages: firstly, we show that when $\|\nabla_t\| \geq \epsilon$, the function descends as fast as the deterministic case; Secondly, when $\|\nabla f(x_t)\| \leq \epsilon$, and $\lambda_{min}(\nabla^2 f(x_t)) \leq -\sqrt{L_H \epsilon}$, i.e., $x_t$ is a saddle point, by a coupling argument it is shown that either the function descends or the sequence of iterates are stuck around the saddle point. But then it is shown that the stuck region is narrow enough so that the iterates escape the saddle points with high probability. We now require a condition on the tail of the stochastic gradient.

**Assumption 3.1** *For any $x \in \mathbb{R}^d$, $\mathbb{P}\left( \|\nabla F(x, \xi) - \nabla f(x)\| \geq \tau \right) \leq 2e^{-\frac{\tau^2}{2\mathbb{E}\left[ \|\nabla F(x,\xi) - \nabla f(x)\|^2 \right]}}$.*

Such light-tail conditions are common in the stochastic optimization literature to obtain high-probability bounds; see for example [20, 25]. Note that under Assumption 2.4, Assumption 3.1 is equivalent to

$$\mathbb{P}\left( \|\nabla F(x, \xi) - \nabla f(x)\| \geq \tau \right) \leq 2e^{-\tau^2/(2(\rho-1)\|\nabla f(x)\|^2)} \tag{5}$$

We now present our main result on PSGD.

**Theorem 3.1** *a) Under Assumptions 2.2, 2.3 on the function $f(\cdot)$, and Assumptions 2.4, and 3.1, choosing,*

$$\eta = \log\left(\frac{1}{\epsilon}\right)^{-2} \Big/ a_0 \log\left(\frac{f(x_0) - f^*}{\delta\epsilon}\right), \; r = \epsilon^{1.5}\log(\epsilon^{-1})^{-3}, n_1 = 512c(\rho - 1)\log(\epsilon^{-1}), \tag{6}$$

*with probability at least $1 - \delta$, half of the iterations of Algorithm 1 will be $\epsilon$-local minimizers after $T$ iterations where,*

$$T = a_1 \max\left\{ \frac{(f(x_0) - f^*)\mathcal{T}_1}{\mathcal{F}_1}, \frac{(f(x_0) - f^*)}{\eta\epsilon^2} \right\} = \tilde{\mathcal{O}}\left( \frac{\log\left(\frac{1}{\delta}\right)}{\epsilon^2} \right), \tag{7}$$

*where $a_0$, $a_1$ are constants, and $\mathcal{T}_1 = 0.5\log\left(\frac{1}{\epsilon}\right)^3/\sqrt{\epsilon}$, and $\mathcal{F}_1 = \epsilon^{1.5}/\log\left(\frac{1}{\epsilon}\right)^7$.*
*b) Under Assumptions 2.1, 2.2, 2.3, 2.4, and 3.1 in the zeroth order-setting, choosing,*

$$\eta = \frac{\kappa_0}{\log\left(\frac{f(x_0)-f^*}{\delta\epsilon}\right)} \quad r = \kappa_1\epsilon \quad \nu = \frac{\kappa_4\epsilon}{d\log\left(\frac{1}{\epsilon}\right)} \quad n_1 = \frac{\kappa_5\log\left(\frac{1}{\epsilon}\right)^5 d^{1.5}\sqrt{\rho-1}}{\epsilon^{2.5}} \tag{8}$$

*with probability at least $1 - \delta$, half of the iterations of Algorithm 1 will be $\epsilon$-local minimzers, after $T$ iterations, where,*

$$T = \kappa_9 \max\left\{ \frac{(f(x_0) - f^*)\mathcal{T}_0}{\mathcal{F}_0}, \frac{(f(x_0) - f^*)}{\eta\epsilon^2} \right\} = \tilde{\mathcal{O}}\left( \frac{\log\left(\frac{1}{\delta}\right)}{\epsilon^2} \right). \tag{9}$$

*Here, $\kappa_i, i = 1, 2, \cdots, 9$ are absolute constants, and $\mathcal{T}_0 = \kappa_3 \frac{\log\left(\frac{1}{\epsilon}\right)^2 \log(d)^2}{\sqrt{\epsilon}}$, and $\mathcal{F}_0 = \kappa_8\epsilon^{1.5}$.*
*Hence, the total number of zeroth-order oracle calls is $Tn_1 = \tilde{\mathcal{O}}\left( \frac{d^{1.5}\sqrt{\rho-1}}{\epsilon^{4.5}} \right)$.*

**Remark 2** *Note that the complexity result in (7) for the PSGD in the first-order setting matches corresponding complexity of perturbed gradient descent on deterministic optimization problems.*

**Remark 3** *We briefly highlight on the difficulty associated with proving the result in (9). First note that in the first-order proof, and also in [25], it is assumed that the noise $\xi$ is sub-gaussian. But for the zeroth-order gradient $g_t$ as defined in (3), $\|g_t - \nabla_t\|$ no longer has sub-Gaussian tails. Also note*

that we have from [38], $\mathbf{E}_{u_{t,i}}[g_{t,i}] = \nabla F_\nu(x_t, \xi_{t,i}) = \nabla \mathbf{E}_{u_{t,i}}[F(x + \nu u_{t,i}, \xi_{t,i})]$. So $g_t$ is not an unbiased estimator of $\nabla_t$. But as shown in [38], $\nabla F_\nu(x_t, \xi_{t,i})$ is close to $\nabla F(x_t, \xi_{t,i})$. So we first need to establish concentration properties for $g_t$ in the zeroth-order setting. Towards this, we show that $g_t$ is $\alpha$-sub-exponential with $\alpha = 2/3$, even if $\xi$ is sub-gaussian, i.e., the noise in the gradient estimates has heavier tail (Lemma A.1). This leads to the obtained complexity bounds in (9).

**Remark 4** *Note that $\mathcal{T}_1$ and $\mathcal{T}_0$ are the number of iterations required to descend by $\mathcal{F}_1$ and $\mathcal{F}_0$ respectively in the first and zeroth-order setting, after the algorithm hits a saddle point. As shown in [25], without SGC, $\mathcal{T}_1 = \tilde{O}(\epsilon^{-2.5})$. In this paper we show that, under SGC, $\mathcal{T}_1 = \mathcal{T}_0 = \tilde{O}(\epsilon^{-0.5})$. This shows under SGC, it is indeed possible to escape saddle point faster.*

We highlight here that [4, 18] recently considered escaping saddle points in the zeroth-order setting. However they assume that the function being optimized is deterministic (which means exact gradients could be obtained) and is bounded (which means sub-Gaussian tails are possible for the zeroth-order gradient estimator). These two assumptions are however highly impractical and are not satisfied by several situations in practice where zeroth-order optimization techniques are utilized. To the best of our knowledge, there is no known bound on the number of times zeroth-order oracle should accessed for (9) to hold, when SGC does not hold and only the following standard variance assumption on the unseen stochastic gradient holds (see, e.g., [20]) for some $\sigma > 0$,

$$\mathbf{E}\left[\left\|\frac{1}{n_1}\sum_{i=1}^{n_1}\nabla F(x_t, \xi_i) - \nabla_t\right\|^2\right] \leq \frac{\sigma^2}{n_1}. \tag{10}$$

For completeness we present the corresponding result below, which serves as a reference to compare our results with the SGC assumption to what one could obtain without it.

**Theorem 3.2** *Under Assumptions 2.1, 2.2, 2.3, 2.4, and 3.1, we have the following: In the zeroth order-setting, choosing,*

$$\eta = \frac{\kappa_0}{\log\left(\frac{f(x_0)-f^*}{\delta\epsilon}\right)} \quad r = \kappa_1\epsilon \quad \nu = \frac{\kappa_4\epsilon}{d\log\left(\frac{1}{\epsilon}\right)} \quad n_1 = \frac{\kappa_5\log\left(\frac{1}{\epsilon}\right)^5 d^{1.5}\sigma}{\epsilon^{3.5}} \tag{11}$$

*with probability at least $1 - \delta$, half of the iterations of Algorithm 1 will be $\epsilon$-local minimzers, after $T$ iterations, where,*

$$T = \kappa_9 \max\left\{\frac{(f(x_0)-f^*)\mathcal{T}_0}{\mathcal{F}_0}, \frac{(f(x_0)-f^*)}{\eta\epsilon^2}\right\} = \tilde{\mathcal{O}}\left(\frac{\log\left(\frac{1}{\delta}\right)}{\epsilon^2}\right). \tag{12}$$

*Here, $\kappa_i, i = 1, 2, \cdots, 9$ are absolute constants, and $\mathcal{T}_0 = \kappa_3 \frac{\log\left(\frac{1}{\epsilon}\right)^2 \log(d)^2}{\sqrt{\epsilon}}$ and $\mathcal{F}_0 = \kappa_8\epsilon^{1.5}$. Hence, the total number of zeroth-order oracle calls is $Tn_1 = \tilde{\mathcal{O}}\left(\frac{d^{1.5}\sigma}{\epsilon^{5.5}}\right)$.*

**Remark 5** *A generic reduction was proposed in [2] for using any algorithm that converges to a first-order stationary points at a particular rate, to converge to a local minimizer at the same rate. The results in [2] are not directly applicable to the zeroth-order setting due to their assumptions. However, assuming that their assumption could be relaxed to get it work in the zeroth-order setting, it is interesting to examine if the results in [20] for converging to first-order stationary solution could be combined with the reduction proposed in [2] to establish a result similar to Theorem 3.2. To make the result of [20] hold with the same probability as in Theorem 3.2, we would require $O(d\,\epsilon^{-6})$ calls to the stochastic zeroth-order oracle. Hence, in certain regimes it is plausible we obtain improved results. It is interesting future work to examine this further rigorously.*

## 4 Stochastic Cubic-Regularized Newton's Method

In this section we analyze Cubic-Regularized (CR) Newton method under interpolation regime. In non-interpolation like stochastic setting, CR Newton achieves a rate of $\mathcal{O}(\epsilon^{-3.5})$ as compared to $\mathcal{O}(\epsilon^{-4})$ attained by PSGD. Here we show that CR Newton achieves a rate of $\mathcal{O}(\epsilon^{-2.5})$ under

---
**Algorithm 2** Cubic-Regularized Newton Algorithm
---
**Input:** $x_1 \in \mathbb{R}^d$, $T$, $M$, $n_1$, $n_2$
**for** $t = 1$ to $T$ **do**
**Set** $g_t = \frac{1}{n_1} \sum_{i=1}^{n_1} g_{t,i}$ **where**

$$g_{t,i} = \nabla F\left(x_t, \xi_{t,i}^G\right) \qquad\qquad\qquad\qquad \text{(Higher-order)}$$

$$g_{t,i} = \frac{F(x_t + \nu u_{t,i}^G, \xi_{t,i}^G) - F(x_t, \xi_{t,i}^G)}{\nu} u_i^G \qquad\qquad \text{(Zeroth-order)}$$

**Set** $H_t = \frac{1}{n_2} \sum_{i=1}^{n_2} H_{t,i}$ **where**

$$H_{t,i} = \nabla^2 F\left(x_t, \xi_{t,i}^H\right) \qquad\qquad\qquad\qquad \text{(Higher-order)}$$

$$H_{t,i} = \frac{F(x_t + \nu u_{t,i}^H, \xi_{t,i}^H) + F(x_t - \nu u_{t,i}^H, \xi_{t,i}^H) - 2F(x_t, \xi_{t,i}^H)}{2\nu^2}\left(u_{t,i}^H u_{t,i}^{H\top} - I\right) \quad \text{(Zeroth-order)}$$

**where** $u_{t,i}^{G[H]} \sim \mathcal{N}\left(\mathbf{0}, \boldsymbol{I}_d\right) \forall t = 1, 2, \cdots, T, i = 1, 2, \cdots, n_1[n_2]$
**Update**

$$x_{t+1} = \operatorname*{argmin}_y m_t\left(x_t, y, g_t, H_t, M\right), \qquad\qquad (13)$$

**where**

$$m_t(y) = f(x_t) + (y - x_t)^\top g_t + \frac{1}{2}(y - x_t)^\top H_t(y - x_t) + \frac{M}{6}\|y - x_t\|^3 \qquad (14)$$

**end for**
---

SGC. Even though this rate is better than non-interpolation like stochastic setting, quite interestingly, CR Newton method fails to achieve deterministic rate of $\mathcal{O}(\epsilon^{-1.5})$ unlike PSGD. We believe that without stronger assumption on the Hessian estimator noise as well, CR Newton will perform worse than PSGD. In this section let $\mathcal{F}_t$ be the filtration generated until time $t$, i.e., in the higher-order setting $\mathcal{F}_t = \sigma(\{\xi_{i,j}^G\}_{i,j=1}^{t,n_1}, \{\xi_{i,j}^H\}_{i,j=1}^{t,n_2})$, and in the zeroth-order setting $\mathcal{F}_t = \sigma(\{\xi_{i,j}^G\}_{i,j=1}^{t,n_1}, \{u_{i,j}^G\}_{i,j=1}^{t,n_1}, \{\xi_{i,j}^H\}_{i,j=1}^{t,n_2}, \{u_{i,j}^H\}_{i,j=1}^{t,n_2})$. We now present our main result.

**Theorem 4.1** *Let $f$ be a function for which Assumptions 2.2 and 2.3 are true. Then under SGC, i.e., under Assumption 2.4, for Algorithm 2, we have:*

a) *In the higher-order setting, choosing*
$$T = \frac{144\left(f(x_1) - f^*\right)}{M\epsilon^{\frac{3}{2}}} \ n_1 = \frac{\mu_0(\rho - 1)}{\epsilon} \ n_2 = \epsilon^{-1}, M = \max\left(L_H, \frac{1}{4}, \left(0.004 L_G \epsilon^{\frac{1}{4}} + \sigma_2 \epsilon^{\frac{1}{4}}\right), 40\sigma_2\right)$$
$$(15)$$

*we get,* $\max\left(\sqrt{\frac{\mathbf{E}[\|\nabla f(x_R)\|]}{144 M}}, -\frac{\mathbf{E}[\lambda_{1,R}]}{9M}\right) \leq \sqrt{\epsilon}$, *where $\mu_0$ is a constant independent of $\epsilon$ and $d$, and $R$ is an integer random variable uniformly distributed over the support $\{1, 2, \cdots, T\}$. The total number of first-order and second-order oracle calls are hence $\mathcal{O}\left(\epsilon^{-\frac{5}{2}}\right)$.*

b) *In the zeroth-order setting, choosing*
$$T = \frac{\mu_0\left(f(x_1) - f^*\right)}{M\epsilon^{\frac{3}{2}}}, n_1 = \frac{\mu_1(d + 5)}{\epsilon}, M = \mu_4, \nu = \frac{\mu_3 \epsilon}{(d + 16)^{\frac{5}{2}}}, n_2 = \frac{\mu_2(1 + 2\log 2d)(d + 16)^4}{\epsilon}$$
$$(16)$$

*we get,* $\max\left(\sqrt{\mathbf{E}\left[\|\nabla f\left(x_R\right)\|\right]}, -\mathbf{E}\left[\lambda_{1,R}\right]\right) \leq \mathcal{O}\left(\sqrt{\epsilon}\right)$, *where $\mu_0, \mu_1, \mu_2, \mu_3, \mu_4$ are constants independent of $\epsilon$, and $d$, and $R$ is an integer random variable uniformly distributed over the support $\{1, 2, \cdots, T\}$.. The total number of first-order oracle calls is $\mathcal{O}\left(d/\epsilon^{\frac{5}{2}}\right)$, and the number of second-order oracle calls is $\mathcal{O}\left(d^4 \log d/\epsilon^{\frac{5}{2}}\right)$.*

**Remark 6** *The above results only require Assumption 2.4, which is a gradient-level property of inter-polation condition. As SCRN is a second-order algorithm, an assumption like "if the min eigenvalue of true Hessian at a point is non-negative, then min eigenvalue of stochastic Hessian is almost surely also non-negative" might be required to capture second-order properties of interpolation. Such an assumption could then be used to obtain a result similar to Lemma 2.1 for stochastic Hessians, to improve the rates in Theorem 4.1. Formalizing this intuition is an extremely interesting future work.*

**Remark 7** *In comparison to the PSGD algorithm, we obtain the results for the SCRN algorithm in expectation. We highlight that it is straightforward to obtain to obtain a high-probability result in the higher-order setting. However, it is technically challenging to do so for the zeroth-order setting. This is due to the difficulty associated with obtaining sharper concentration results for the zeroth-order Hessian estimator, which we leave as future work. In Theorem 4.1, we presented the results in expectation for both settings to maintain uniformity of presentation. In Algorithm 2 we assume that the exact solution to (13) is available. We remark that it is possible to relax this assumption following the approach of [47] which in turn leveraged the results in [7] showing that the subproblem in (13) can be solved with high probability using gradient descent.*

## 5   Summary

In this work, we analyze the oracle complexity of two standard algorithms –the perturbed stochastic gradient descent algorithm and the stochastic cubic-regularized Newton's method–for escaping saddle-points in nonconvex stochastic optimization. We show that under interpolation-like conditions satisfied in modern over-parametrized machine learning problems, PSGD and SCRN obtain improved rates for escaping saddle-points. In particular the above stated improvements are obtained for the vanilla versions of PSGD and SCRN algorithms and are not based on any complicated variance reduction techniques. For future work, it is extremely interesting to bridge the gap between SCRN and its deterministic counterpart. The key to this is come up with a Hessian-based interpolation-like assumption, which is both practically meaningful and theoretically sound.

## Broader Impact

We focus in this work on establishing theoretical justification for a practically observed phenomenon: Stochastic gradient method and its relatives perform well for training deep neural networks with complicated nonconvex landscape. The result presented will benefit researchers and practitioners who are interested in understanding the theoretical underpinnings of stochastic optimization for deep learning. Although our work in this draft is theoretical, it might have a positive impact for various practical applications of neural networks.

## Funding Disclosure

The AR was supported in parts by the NSF Grant CCF-1934568. The research of KB was supported in parts by UC Davis CeDAR (Center for Data Science and Artificial Intelligence Research) Innovative Data Science Seed Funding Program. The research of PM was sponsored in part by the U.S. Army Combat Capabilities Development Command Army Research Laboratory and was accomplished under Cooperative Agreement Number W911NF-13-2-0045 (ARL Cyber Security CRA). The views and conclusions contained in this document are those of the authors and should not be interpreted as representing the official policies, either expressed or implied, of the Combat Capabilities Development Command Army Research Laboratory or the U.S. Government. The U.S. Government is authorized to reproduce and distribute reprints for Government purposes notwithstanding any copyright notation here on.

## Footnotes

[1]Here, $\tilde{\mathcal{O}}$ hides log factors.

[2]It is possible to obtain $\tilde{\mathcal{O}}(1/\epsilon^{11.75})$ complexity using accelerated method in deterministic setting; see [26].

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
