[Supplementary Material]

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

}\left[F(x,\xi)\right] = f(x)$, $\mathbf{E}\left[\nabla F(x,\xi)\right] = \nabla f(x)$, $\mathbf{E}\left[\nabla^2 F(x,\xi)\right] = \nabla^2 f(x)$, and $\mathbf{E}\left[\|\nabla^2 F(x,\xi) - \nabla^2 f(x)\|_F^4\right] \leq \sigma_2^4$, where $\|\cdot\|_F$ is the Frobenius norm.*

**Assumption 2.6 (Higher-order oracles)** *For any $x \in \mathbb{R}^d$, (i) the first-order oracle outputs an estimate $\nabla F(x,\xi)$ of $\nabla f(x)$ such that $\mathbf{E}\left[\nabla F(x,\xi)\right] = \nabla f(x)$ and (ii) the second-order oracle, in addition outputs an estimate $\nabla^2 F(x,\xi)$ of $\nabla^2 f(x)$ such that, $\mathbf{E}\left[\nabla^2 F(x,\xi)\right] = \nabla^2 f(x)$, and $\mathbf{E}\left[\|\nabla^2 F(x,\xi) - \nabla^2 f(x)\|_F^4\right] \leq \sigma_2^4$.*

Such assumptions on the zeroth-order and higher-order oracles are standard in the literature; see for example [20, 38**?** ]. Our goal in this paper is to reach an approximate local minimizer (also called as a second-order stationary point) of a non-convex function, which is defined as follows:

**Definition 2.1 ($\epsilon$-Local Minimizer)** *Let Assumption 2.3 hold for a function $f$. Then a point $\bar{x}$ is called a $\epsilon$-second-order stationary point if,*

$$\max\left(\sqrt{\|\nabla f(\bar{x})\|}, -\frac{\lambda_{min}\left(\nabla^2 f(\bar{x})\right)}{L_H}\right) \leq \sqrt{\epsilon} \tag{2}$$

*where $\lambda_{min}\left(\nabla^2 f(\bar{x})\right)$ is the minimum eigenvalue of $\nabla^2 f(\bar{x})$.*

---

**Algorithm 1** Perturbed Stochastic Gradient Descent Algorithm

---

**Input:** $x_0 \in \mathbb{R}^d$, $\eta$, $r$.
**for** $t = 0$ to $T$ **do**
**Set** $g_t = \frac{1}{n_1} \sum_{i=1}^{n_1} g_{t,i}$ **where**

$$g_{t,i} = \nabla F\left(x_t, \xi_{t,i}\right) \qquad\qquad \text{(First-order)}$$

$$g_{t,i} = \frac{F(x_t + \nu u_{t,i}, \xi_{t,i}) - F(x_t, \xi_{t,i})}{\nu} u_i \qquad\qquad \text{(Zeroth-order)}$$

**and** $u_{t,i} \sim \mathcal{N}\left(\mathbf{0}, \boldsymbol{I}_d\right) \forall t = 1, 2, \cdots, T, i = 1, 2, \cdots, n_1$
**Sample** $\theta_t \in \mathcal{N}\left(\mathbf{0}, r^2 \boldsymbol{I}_d\right)$
**Update** $x_{t+1} = x_t - \eta\left(g_t + \theta_t\right)$

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

# A   Proof of Theorem 3.1.

**Preliminaries I:** We first present preliminary results regarding the zeroth-order setting.

**Lemma A.1** *Let Assumption 2.2, and 3.1 be true for $F$. Then, in the zeroth-order setting, $\|\zeta_t\|$ is a $2/3$-sub-exponential variable. i.e.,*

$$\mathbb{P}\left(\|\zeta_t\| \geq \tau\right) \leq 4d\exp\left(-K_1 \min\left[\left(\frac{\sqrt{n_1}\tau'}{\Upsilon_t\sqrt{d}}\right)^2, \left(\frac{n_1\tau'}{\Upsilon_t\sqrt{d}}\right)^{2/3}\right]\right), \tag{17}$$

*where $\tau' = \tau - \frac{\nu}{2}L_G(d+3)^{\frac{3}{2}}$, and $\Upsilon_t = \frac{\nu L_G(d+2)}{2} + c_0\sqrt{(\rho-1)(d+1)}\|\nabla_t\|$.*

We will choose $n_1$ such that we have $(n_1\tau'/(\Upsilon_t\sqrt{d}))^{2/3} \leq (\sqrt{n_1}\tau'/(\Upsilon_t\sqrt{d}))^2$. So from now on we will only consider the heavier subexponential tail.

**Lemma A.2** *Let Assumption 2.2, and 3.1 be true for $F$. Then, in the zeroth-order setting*

$$\mathbf{E}\left[\exp(s(c_t\|\zeta_t\|)^{\frac{1}{3}})\right] \leq 9d\exp(s^2/b_{1,t}),$$

*where $s > 0$, $b_{1,t} = b_{0,t}/c_t^{2/3}$, and $b_{0,t} = K_1 n_1^{2/3}/(\Upsilon_t\sqrt{d})^{2/3}$.*

**Lemma A.3** *Let Assumption 2.2, and 3.1 be true for $F$. Then, in the zeroth-order setting for $l > 0$, with probability at least $1 - e^{-l}$ we have*

$$\eta\sum_{i=0}^{t-1}\nabla_i^\top\zeta_i \leq \frac{8\eta\sqrt{dt}}{K_1^{\frac{3}{2}}n_1}(t\log 9d + l)^{\frac{3}{2}}\sum_{i=0}^{t-1}\left(\frac{\nu L_G(d+2)}{2}\|\nabla_i\| + C_0\sqrt{(\rho-1)(d+1)}\|\nabla_i\|^2\right).$$

**Lemma A.4** *Let Assumption 2.2, and 3.1 be true for $F$. Then, for $l > 0$, with probability at least $1 - e^{-l}$ we have*

$$\sum_{i=0}^{t-1}\|\zeta_i\|^2 \leq \frac{128dt^2(t\log 9d + l)^3}{K_1^3 n_1^2}\sum_{i=0}^{t-1}\left(\left(\frac{\nu L_G(d+2)}{2}\right)^2 + C_0^2(\rho-1)(d+1)\|\nabla_i\|^2\right).$$

**Preliminaries II:** We next present preliminary results regarding the iterates of PSGD. First, we show that the effect of PSGD updates comprises of two parts - the first term on the RHS of (19), and (22) represent the decrease in the function values, and the rest of the terms on the RHS represent possible increase in function value due to noise in the gradient estimator and introduced perturbation.

**Lemma A.5** *Under Assumption 2.1, 2.2, 2.3, 2.4 and 3.1, for any fixed $\mathcal{T}_0, \mathcal{T}_1, l > \log 4$, with probability at least $1 - 4e^{-l}$, for Algorithm 1 we get*

    *a) for the first-order setting, choosing*

$$n_1 \geq 512lc(\rho-1) \quad \eta \leq \frac{32lc}{3L_G(l+c)} \tag{18}$$

    *we have*

$$f(x_{\mathcal{T}_1}) - f(x_0) \leq -\frac{\eta}{16}\sum_{i=0}^{\mathcal{T}_1}\|\nabla_i\|^2 + 3c\eta^2 r^2(\mathcal{T}_1 + l)L_G + 32cl\eta r^2 \tag{19}$$

    *b) for the zeroth-order case, selecting parameters such that*

$$\frac{384L_G C_0^2 d(\rho-1)(d+1)\mathcal{T}_0^2(\mathcal{T}_0\log 9d + l)^3}{K_1^3 n_1^2} \leq \frac{1}{16} \tag{20}$$

$$\frac{8C_0\sqrt{(\rho-1)d(d+1)\mathcal{T}_0}}{K_1^{\frac{3}{2}}n_1}(\mathcal{T}_0\log 9d + l)^{\frac{3}{2}} \leq \frac{1}{16} \tag{21}$$

*we have*

$$f(x_{\mathcal{T}_0}) - f(x_0) \leq -\frac{\eta}{16} \sum_{i=0}^{\mathcal{T}_0-1} \|\nabla_i\|^2 + \wp(r, l, \nu, \eta, d, \mathcal{T}_0) \tag{22}$$

*where*

$$\wp(r, l, \nu, \eta, d, \mathcal{T}_0) = 16cl\eta r^2 + 3cL_G\eta^2 r^2(\mathcal{T}_0 + l)$$

$$+ \frac{8\nu\eta LL_G(d+2)\sqrt{d}\mathcal{T}_0^{\frac{3}{2}}}{2K_1^{\frac{3}{2}}n_1}(\mathcal{T}_0 \log 9d + l)^{\frac{3}{2}} + \frac{96L_G^3\nu^2\eta^2 d(d+2)^2\mathcal{T}_0^3(\mathcal{T}_0 \log 9d + l)^3}{K_1^3 n_1^2}$$

In the following Lemma we show that when the function descent is small the iterates move only in a small region.

**Lemma A.6** *Under conditions of Lemma A.5, Algorithm 1 satisfies*

*a) for first-order setting, with probability at least $1 - 8d\mathcal{T}_1 e^{-l}$, for all $\tau \leq \mathcal{T}_1$*

$$\|x_\tau - x_0\|^2 \leq 32\eta\left(\mathcal{T}_1 + 2cl\frac{\rho-1}{n_1}\right)\left(f(x_0) - f(x_{\mathcal{T}_1}) + 3c\eta^2 r^2(\mathcal{T}_1 + l)L_G + 32cl\eta r^2\right) + 4cl\mathcal{T}_1\eta^2 r^2 \tag{23}$$

*b) for zeroth-order setting, with probability at least $1 - 3d\mathcal{T}_0 e^{-l}$, for all $\tau \leq \mathcal{T}_0$*

$$\|x_\tau - x_0\|^2 \leq \eta\mathcal{T}_0\left(32 + \frac{16}{3L_G}\right)(f(x_0) - f(x_{\mathcal{T}_0}) + \wp(r, l, \nu, \eta, d, \mathcal{T}_0)) + 4cl\mathcal{T}_0\eta^2 r^2$$

$$+ \frac{L_G\eta^2\mathcal{T}_0^2\nu^2(d+2)^2}{48C_0^2(\rho-1)(d+1)} \tag{24}$$

We also require the following definition from [25], to proceed.

**Definition A.1** *[25] Let $e_1$ be the eigen-vector corresponding to the minimum eigen-value of $\mathcal{H} = \nabla^2 f(x_0)$, and $\gamma := \lambda_{min}(\nabla^2 f(x_0))$. Also let $\mathcal{P}_{-1}$ be the projection on to the complement subspace of $e_1$. Consider sequences $x_t$, and $x_t'$ that are obtained as separate versions of Algorithm 1, both starting from $x_0$. They are coupled in the first-order (zero-order) setting if both sequences are generated by the same $\mathcal{P}_{-1}\theta_\tau$, and $\xi_\tau$ $(\{\xi_\tau, \{u_{\tau,i}\}_{i=1}^{n_1}\})$, while in $e_1$ direction we have $e_1^\top \theta_\tau = -e_1^\top \theta_\tau'$.*

We next state some intermediate results in Lemma A.7–A.10, to prove in Lemma A.11 that starting from a saddle-point PSGD should either descend or the iterates will be stuck around the saddle point. Then in Lemma A.12 we will show that the stuck region is narrow enough so that the iterates will escape and consequently the function will have sufficient descent.

**Lemma A.7** *[25] Consider the coupling sequences $x_\tau$ and $x_\tau'$ as in Definition A.1 and let $\hat{x}_\tau = x_\tau - x_\tau'$. Then $\hat{x}_t = -q_h(t) - q_{sg}(t) - q_p(t)$, where:*

$$q_h(t) := \eta\sum_{\tau=0}^{t-1}(I - \eta\mathcal{H})^{t-1-\tau}\Delta_\tau\hat{x}_\tau, \quad q_{sg}(t) := \eta\sum_{\tau=0}^{t-1}(I - \eta\mathcal{H})^{t-1-\tau}\hat{\zeta}_\tau, \quad q_p(t) := \eta\sum_{\tau=0}^{t-1}(I - \eta\mathcal{H})^{t-1-\tau}\hat{\theta}_\tau$$

*where $\Delta_t := \int_0^1 (\nabla^2 f(\phi x_t + (1-\phi)x_t')d\phi - \mathcal{H}$, and $\hat{\zeta}_\tau := \zeta_\tau - \zeta_\tau'$, $\hat{\theta}_\tau = \theta_\tau - \theta_\tau'$.*

**Lemma A.8** *[25] Denote $\alpha(t) := \left[\sum_{i=0}^{t-1}(1 + \eta\gamma)^{2(t-1-\tau)}\right]^{\frac{1}{2}}$, and $\beta(t) = (1 + \eta\gamma)^t/\sqrt{2\eta\gamma}$. If $\eta\gamma \in [0, 1]$, then $(1)\alpha(t)\beta(t)$ for any $t \in \mathbb{N}$; and $(2)$ $\alpha(t) \geq \beta(t)/\sqrt{3}$ for $t \geq \ln(2)/(\eta\gamma)$.*

**Lemma A.9** *[25] Under the notation of Lemma A.7, and A.8, we have $\forall t > 0$:*

$$\mathbb{P}\left(\|q_p(t)\| \leq \frac{c\beta(t)\eta r}{\sqrt{d}}\sqrt{l}\right) \geq 1 - 2e^{-l}$$

$$\mathbb{P}\left(\|q_p(\mathcal{T}_{1[0]})\| \geq \frac{\beta(\mathcal{T}_{1[0]})\eta r}{10\sqrt{d}}\right) \geq \frac{2}{3}$$

*We use $1[0]$ to denote that the inequality holds for both subscripts $1$ and $0$.*

**Lemma A.10** *Under the notation of Lemma A.7 and A.8, if*

$$\eta \mathcal{S}\mathcal{T}_{1[0]} \max(L_H, L_G) \leq \frac{1}{l} \quad c \leq \sqrt{l}/40 \tag{25}$$

a) *[25]then in the first-order case, we have*

$$\mathcal{P}\left(\min\{f(x_{\mathcal{T}_1}) - f(x_0), f(x'_{\mathcal{T}_1}) - f(x_0)\} \leq -\mathcal{F}_1, \text{ or } \forall t \leq \mathcal{T}_1 : \|q_h(t) + q_{sg}(t)\| \leq \frac{\beta(t)\eta t}{20\sqrt{d}}\right)$$
$$\geq 1 - 10d\mathcal{T}_1{}^2 \log\left(\frac{\mathcal{S}_1\sqrt{d}}{\eta r}\right)e^{-l}$$

b) *in the zeroth-order case, we have*

$$\mathcal{P}\left(\min\{f(x_{\mathcal{T}_0}) - f(x_0), f(x'_{\mathcal{T}_0}) - f(x_0)\} \leq -\mathcal{F}_0, \text{ or } \forall t \leq \mathcal{T}_0 : \|q_h(t) + q_{sg}(t)\| \leq \frac{\beta(t)\eta t}{20\sqrt{d}}\right)$$
$$\geq 1 - 3\mathcal{T}_0{}^2 e^{-l}$$

**Lemma A.11**     a) *[25] Under the setting of Lemma A.5, for the first-order setting, we have*

$$\mathbb{P}\left(\min\{f(x_{\mathcal{T}_1}) - f(x_0), f(x'_{\mathcal{T}_1}) - f(x_0)\} \leq -\mathcal{F}_1, \text{ or } \forall t \leq \mathcal{T}_1 : \max\{\|x_t - x_0\|^2, \|x'_t - x_0\|^2\} \leq \mathcal{S}_1^2\right)$$
$$\geq 1 - 16d\mathcal{T}_1 e^{-l}$$

b) *for the zeroth-order setting, we have*

$$\mathbb{P}\left(\min\{f(x_{\mathcal{T}_0}) - f(x_0), f(x'_{\mathcal{T}_0}) - f(x_0)\} \leq -\mathcal{F}_0, \text{ or } \forall t \leq \mathcal{T}_0 : \max\{\|x_t - x_0\|^2, \|x'_t - x_0\|^2\} \leq \mathcal{S}_0^2\right)$$
$$\geq 1 - 4d\mathcal{T}_0 e^{-l}$$

In the following Lemma we show that while escaping from a saddle point, the PSGD descends more than it ascends with high probability.

**Lemma A.12** *Let Under Assumption 2.1, 2.2, 2.3, 2.4, and 3.1 are true. Under condition (25), for any fixed $t_0 > 0$, let $x_0$ satisfies*

$$\|\nabla_0\| \leq \epsilon \quad \lambda_{min}(\nabla^2 f(x_0)) \leq -\sqrt{L_H \epsilon}.$$

*Then*

a) *if $\eta, r, n_1$ are chosen as in (6), $\mathcal{T}_1 = 0.5\log\left(\frac{1}{\epsilon}\right)^3 / \sqrt{\epsilon}$, $\mathcal{F}_1 = \epsilon^{1.5} / \log\left(\frac{1}{\epsilon}\right)^7$, $\mathcal{S}_1 = \frac{\sqrt{\epsilon}}{\log\left(\frac{1}{\epsilon}\right)^2}$, $l = a_0 \log\left(\frac{f(x_0) - f^*}{\delta\epsilon}\right)$, then the sequence generated by Algorithm 1 in the first-order case satisfies*

$$\mathbb{P}\left(f(x_{t_0+\mathcal{T}_1}) - f(x_{t_0}) \leq 0.1\mathcal{F}_1\right) \geq 1 - 4e^{-l} \qquad \text{and} \tag{26}$$
$$\mathbb{P}\left(f(x_{t_0+\mathcal{T}_1}) - f(x_{t_0}) \leq -\mathcal{F}_1\right) \geq \frac{1}{3} - 9d\mathcal{T}_1{}^2 \log\left(\frac{\mathcal{S}_1\sqrt{d}}{\eta r}\right)e^{-l} \tag{27}$$

b) *if $\eta, r, n_1$ are chosen as in (8), $\mathcal{T}_0 = \kappa_3 \frac{\log\left(\frac{1}{\epsilon}\right)^2 \log(d)^2}{\sqrt{\epsilon}}$, $\mathcal{F}_0 = \kappa_8 \epsilon^{1.5}$, $\mathcal{S}_0 = \frac{\kappa_7 \sqrt{\epsilon}}{\log\left(\frac{1}{\epsilon}\right)^2}$ and $l = \kappa_6 \log\left(\frac{d(f(x_0) - f^*)}{\delta\epsilon}\right)$, then the sequence generated by Algorithm 1 in the zeroth-order case satisfies*

$$\mathbb{P}\left(f(x_{t_0+\mathcal{T}_0}) - f(x_{t_0}) \leq 0.1\mathcal{F}_0\right) \geq 1 - 4e^{-l} \qquad \text{and} \tag{28}$$
$$\mathbb{P}\left(f(x_{t_0+\mathcal{T}_0}) - f(x_{t_0}) \leq -\mathcal{F}_0\right) \geq \frac{1}{3} - \frac{3}{2}\mathcal{T}_0{}^2 e^{-l} \tag{29}$$

**Finishing the proof:** By combining the above results, we prove Theorem 3.1. The proof is divided in two parts – in the first part we show that the function descends enough when the gradient is large and in the second part we show that the iterates do escape from the saddle points and then function has sufficient descent.

**Choice of parameters for Zeroth-order case.** As the expressions involved in the analysis of the zeroth order case are little complicated, we show explicitly here how to choose the parameters. First define,

$$\Xi := \frac{32\sqrt{d(\mathcal{T}_0+1)}\eta\beta(\mathcal{T}_0+1)((\mathcal{T}_0+1)\log 9d + \log 2 + l)^{\frac{3}{2}}}{K_1^{3/2}n_1} \tag{30}$$

The choice of the parameters should be such that the following equations are satisfied:

$$\frac{384 L_G C_0^2 d(\rho-1)(d+1)\mathcal{T}_0{}^2(\mathcal{T}_0\log 9d + l)^3}{K_1^3 n_1^2} \leq \frac{1}{16},$$

$$\frac{8C_0\sqrt{(\rho-1)d(d+1)\mathcal{T}_0}}{K_1^{\frac{3}{2}}n_1}(\mathcal{T}_0\log 9d + l)^{\frac{3}{2}} \leq \frac{1}{16},$$

$$\eta\mathcal{S}_0\mathcal{T}_0\max(L_H, L_G) \leq \frac{1}{l}, \quad c \leq \sqrt{l}/40,$$

$$\Xi \cdot \sum_{i=0}^{\mathcal{T}_0}\left(\frac{\nu L_G(d+2)}{2} + C_0\sqrt{(\rho-1)(d+1)}L\right) \leq \frac{\beta(\mathcal{T}_0)r}{40\sqrt{d}},$$

$$\frac{(1+\eta\gamma)^{\mathcal{T}_0}\sqrt{\eta}r}{40\sqrt{2\gamma d}} > \mathcal{S}_0, \quad \wp(r, l, \nu, \eta, d, \mathcal{T}_0) \leq 0.1\mathcal{F}_0.$$

Furthermore, we need to ensure the RHS of (24) is of the same order of $\mathcal{S}_0^2$.

**Proof** [Proof of Theorem 3.1]

a)
1. First we look at the time instants where $\|\nabla_t\| \geq \epsilon$. If there are more than $\frac{T}{4}$ such time steps, then using Lemma A.5 we have, with probability at least $1 - 4e^{-l}$

$$f(x_T) - f(x_0) \leq -\frac{T\epsilon^2}{64\log\left(\frac{1}{\epsilon}\right)^2} + 3cL_G\frac{\epsilon^3}{\log\left(\frac{1}{\epsilon}\right)^{10}}\left(\frac{0.5\log\left(\frac{1}{\epsilon}\right)^3}{\sqrt{\epsilon}} + \log\left(\frac{1}{\epsilon}\right)\right) + 32c\frac{\epsilon^3}{\log\left(\frac{1}{\epsilon}\right)^7}$$

$$\leq -\frac{T\epsilon^2}{128\log\left(\frac{1}{\epsilon}\right)^2}$$

Letting $T$ as in (7), we get $f(x_T) \leq f(x_0) - T\epsilon^2/128\log\left(\frac{1}{\epsilon}\right)^2 < f^*$ which is impossible.

2. As follows from Claim 2 in the proof of Theorem 16 of [25], we have, with probability at least $1 - 10d\mathcal{T}_0{}^2 T^2\log(\mathcal{S}_1\sqrt{d}/(\eta r))e^{-l}$

$$f(x_T) - f(x_0) \leq -0.1\frac{T\mathcal{F}_1}{\mathcal{T}_1}$$

which implies $f(x_T) \leq f(x_0) - 0.1T\mathcal{F}_1/\mathcal{T}_1 < f^*$ which is impossible.

b)
1. First we look at the time instants where $\|\nabla_t\| \geq \epsilon$. If the parameters are chosen as in (8), $\mathcal{T}_0 = \kappa_3\frac{\log\left(\frac{1}{\epsilon}\right)^2\log(d)^2}{\sqrt{\epsilon}}$, and $l = \kappa_6\log\left(\frac{d(f(x_0)-f^*)}{\delta\epsilon}\right)$ then we have,

$$\wp(r, l, \nu, \eta, d, \mathcal{T}_0) = \mathcal{O}\left(\epsilon^{1.5}\right)$$

If there are more than $\frac{T}{4}$ such time steps, then using Lemma A.5 we have, with probability at least $1 - 4e^{-l}$

$$f(x_T) - f(x_0) \leq -\frac{\kappa_0 T\epsilon^{2.5}}{64\log\left(\frac{1}{\epsilon}\right)} + \mathcal{O}\left(\epsilon^{1.5}\right) \leq -\frac{\kappa_0 T\epsilon^{2.5}}{128\log\left(\frac{1}{\epsilon}\right)}$$

Letting $T$ as in (9), $\kappa_9 \geq 128$, and $\kappa_0\kappa_3/\kappa_8 \geq 128$ we get $f(x_T) \leq f(x_0) - \frac{\kappa_0 T\epsilon^{2.5}}{128\log\left(\frac{1}{\epsilon}\right)} < f^*$ which is impossible.

2. As follows from Claim 2 in the proof of Theorem 16 of [25], we have, with probability at least $1 - 3\mathcal{T}_0^2 T^2 e^{-l}$

$$f(x_T) - f(x_0) \leq -0.1 \frac{T\mathcal{F}_0}{\mathcal{T}_0}$$

which implies $f(x_T) \leq f(x_0) - 0.1 T\mathcal{F}_0/\mathcal{T}_0 < f^*$ when $\kappa_9 \geq 128$ and $T$ is as in (8), which is impossible.

∎

**Proof** [Proof of Theorem 3.2] The proof of Theorem 3.2 is same as Theorem 3.1 except for the concentration properties of $\|\zeta_t\|$. In this case we have $\|\zeta_t\|$ to be $\alpha$-sub-exponential with coefficient $(\Upsilon_t \sqrt{d}/n_1)^{2/3}$ where

$$\Upsilon_t = \frac{\nu L_G(d+2)}{2} + C_0(\sigma + \|\nabla f(x_t)\|)\sqrt{d+1}.$$

So there is an extra term $C_0 \sigma \sqrt{d+1}$ which can neither be made smaller using $\nu$ nor is of the same order as $\nabla f(x_t)$ so that it can be subsumed in other terms involving $\nabla f(x_t)$. Hence, the only way to make the coefficient smaller, which is essential in the proof, is to increase $n_1$. This is main reason why the rate deteriorates in the absence if SGC. For the sake of completeness, we provide below the set of conditions that need to be satisfied to pick the parameters in this setting, below.

**Choice of parameters for Zeroth-order case when SGC does not hold.** When SGC does not hold in the zeroth-order setting the conditions to be satisfied are:

$$\frac{384 L_G C_0^2 d(\rho-1)(d+1)\mathcal{T}_0^2 (\mathcal{T}_0 \log 9d + l)^3}{K_1^3 n_1^2} \leq \frac{\epsilon^2}{16},$$

$$\frac{8 C_0 \sqrt{(\rho-1)d(d+1)\mathcal{T}_0}}{K_1^{\frac{3}{2}} n_1} (\mathcal{T}_0 \log 9d + l)^{\frac{3}{2}} \leq \frac{\epsilon}{16},$$

$$\eta \mathcal{S}_0 \mathcal{T}_0 \max(L_H, L_G) \leq \frac{1}{l}, \quad c \leq \sqrt{l}/40,$$

$$\Xi \cdot \sum_{i=0}^{\mathcal{T}_0} \left( \frac{\nu L_G(d+2)}{2} + C_0 \sqrt{(\rho-1)(d+1)} L \right) \leq \frac{\beta(\mathcal{T}_0)r}{40\sqrt{d}},$$

$$\frac{(1+\eta\gamma)^{\mathcal{T}_0} \sqrt{\eta} r}{40\sqrt{2\gamma d}} > \mathcal{S}_0, \quad \wp(r, l, \nu, \eta, d, \mathcal{T}_0) \leq 0.1\mathcal{F}_0.$$

Furthermore, we need to ensure the RHS of (24) is of the same order of $\mathcal{S}_0^2$. ∎

## A.1 Proofs of Lemmas related to Perturbed Stochastic Gradient Descent

**Assumption A.1** *[25] Consider random vectors $X_1, X_2, \cdots, X_n \in \mathbb{R}^d$, and the corresponding filtrations $\mathcal{F}_i = \sigma(X_1, X_2, \cdots, X_i)$ for $i = 1, 2, \cdots, n$, such that $X_i|\mathcal{F}_{i-1}$ is zero-mean nSG($\sigma_i$) with $\sigma_i \in \mathcal{F}_{i-1}$. That is,*

$$\mathbf{E}[X_i|\mathcal{F}_{i-1}] = 0, \quad P(\|X_i\| \geq t|\mathcal{F}_{i-1}) \leq e^{-\frac{t^2}{2\sigma_i^2}}, \quad \forall t \in \mathbb{R}, \forall i = 1, 2, \cdots, n.$$

**Lemma A.13** *[25] Let $X_1, X_2, \cdots, X_n \in \mathbb{R}^d$ satisfy Assumption 3.1. $u_i \in \mathcal{F}_{i-1}$ be a random vector for $i = 1, 2, \cdots, n$. Then for any $l > 0$, $\lambda > 0$, there exists absolute constant $c$ such that, with probability at least $1 - e^{-l}$:*

$$\sum_i u_i^\top X_i \leq c\lambda \sum_i \|u_i\|^2 \sigma_i^2 + \frac{l}{\lambda}$$

**Lemma A.14** *[25] Let $X_1, X_2, \cdots, X_n \in \mathbb{R}^d$ satisfy Assumption 3.1 with $\sigma_1 = \sigma_2 = \cdots = \sigma_n = \sigma$. Then for any $l > 0$, $\lambda > 0$, there exists absolute constant $c$ such that, with probability at least $1 - e^{-l}$:*

$$\sum_i \|X_i\|^2 \leq c\sigma^2(n+l)$$

**Lemma A.15** *[25] Let $X_1, X_2, \cdots, X_n \in \mathbb{R}^d$ satisfy Assumption 3.1 with fixed $\{\sigma_i\}$ then for any $l > 0$, there exists an absolute coonstant $c$ such that, with probability at least $1 - 2de^{-l}$:*

$$\|\sum_{i=1}^n X_i\| \le c\sqrt{\sum_{i=1}^n \sigma_i^2 l}$$

Let $F_\nu(x,\xi) = \mathbf{E}_u[F(x + \nu u, \xi)]$, and $g_{t,i}^j$, and $\nabla F_\nu(x_t, \xi_i)^j$ denote the $j$-th coordinate of the vector $g_{t,i} = \frac{F(x_t + \nu u_i, \xi_i) - F(x_t, \xi_i)}{\nu} u_i$, and $\nabla F_\nu(x_t, \xi_i)$ respectively.

**Lemma A.16** *[38] Let Assumption 2.2 be true for $F$. Then*

$$\|\nabla F_\nu(x,\xi) - \nabla F(x,\xi)\| \le \frac{\nu}{2} L_G (d+3)^{\frac{3}{2}}$$

**Lemma A.17** *[38] For a Gaussian random vector $u \sim N(0, I_d)$, we have*

$$\mathbf{E}\left[\|u\|^k\right] \le (d+k)^{\frac{k}{2}}$$

**Lemma A.18** *[44] Let $(X_i, Y_i)$, $i = 1, 2, \cdots, n$ be $n$ independent copies of random variables $X$ and $Y$. Let $X$ be a sub-Gaussian random variable with sub-gaussian norm $\|X\|_{\psi_2} \le \Upsilon_1$, and $Y$ be a sub-exponential random variable with sub-exponential norm $\|Y\|_{\psi_1} \le \Upsilon_2$ for some constants $\Upsilon_1$ and $\Upsilon_2$. Then for any $t \ge K \max\left(\Upsilon_1, \Upsilon_1^3\right) \Upsilon_2$ we have*

$$\mathbb{P}\left(|\sum_{i=1}^n X_i Y_i - \mathbf{E}[XY]| \ge t\right) \le 4\exp\left(-K_1 \min\left[\left(\frac{t}{\sqrt{n}\Upsilon_1\Upsilon_2}\right)^2, \left(\frac{t}{\Upsilon_1\Upsilon_2}\right)^{2/3}\right]\right)$$

*where $K$ and $K_1$ are absolute constants.*

**Proof** [Proof of Lemma A.1] Let us write $g_{t,i} = \phi(\nu, u_i, \xi_i) u_i$ where $\phi(\nu, u_i, \xi_i) = \frac{F(x_t + \nu u_i, \xi_i) - F(x_t, \xi_i)}{\nu}$. We will show that $\phi(\nu, u_i, \xi_i)$ is a sub-exponential random variable by showing that its sub-exponential norm or $\psi_1$-norm, defined as $\|.\|_{\psi_1} = \sup_{p \ge 1} p^{-1} \mathbf{E}[|.|^p]^{p^{-1}}$, is finite.

$$\|\phi(\nu, u_i, \xi_i)\|_{\psi_1} = \sup_{p \ge 1} \frac{1}{p} \mathbf{E}[|\phi(\nu, u_i, \xi_i)|^p]^{\frac{1}{p}} = \sup_{p \ge 1} \frac{1}{p} \mathbf{E}_{\xi_i}[\mathbf{E}_{u_i}[|\phi(\nu, u_i, \xi_i)|^p]]^{\frac{1}{p}} \quad (31)$$

We first concentrate on the term $\mathbf{E}_{u_i}[|\phi(\nu, u_i, \xi_i)|^p]$.

$$\mathbf{E}_{u_i}[|\phi(\nu, u_i, \xi_i)|^p] = \mathbf{E}_{u_i}\left[\left|\frac{F(x_t + \nu u_i, \xi_i) - F(x_t, \xi_i) - \nu \nabla F(x_t, \xi_i)^\top u_i}{\nu} + \nabla F(x_t, \xi_i)^\top u_i\right|^p\right]$$

By Minkowski's inequality,

$$\mathbf{E}_{u_i}[|\phi(\nu, u_i, \xi_i)|^p]$$
$$\le \left[\mathbf{E}_{u_i}\left[\left|\frac{F(x_t + \nu u_i, \xi_i) - F(x_t, \xi_i) - \nu \nabla F(x_t, \xi_i)^\top u_i}{\nu}\right|^p\right]^{\frac{1}{p}} + \mathbf{E}_{u_i}\left[|\nabla F(x_t, \xi_i)^\top u_i|^p\right]^{\frac{1}{p}}\right]^p$$
$$\le \left[\frac{\nu L_G}{2} \mathbf{E}_{u_i}[\|u_i\|^{2p}]^{\frac{1}{p}} + \|\nabla F(x_t, \xi_i)\| \mathbf{E}_{u_i}[\|u_i\|^p]^{\frac{1}{p}}\right]^p$$

Using Lemma A.17,

$$\mathbf{E}_{u_i}[|\phi(\nu, u_i, \xi_i)|^p] \le \left[\frac{\nu L_G (d+2p)}{2} + \sqrt{d+p}\|\nabla F(x_t, \xi_i)\|\right]^p$$

Now from (31), using Minkowski's inequality, we get

$$\|\phi(\nu, u_i, \xi_i)\|_{\psi_1} \le \sup_{p \ge 1} \frac{1}{p} \mathbf{E}_{\xi_i}\left[\left(\frac{\nu L_G (d+2p)}{2}\right)^p\right]^{\frac{1}{p}} + \sup_{p \ge 1} \frac{1}{p} \mathbf{E}_{\xi_i}\left[\left(\sqrt{d+p}\|\nabla F(x_t, \xi_i)\|\right)^p\right]^{\frac{1}{p}}$$

$$\leq \frac{\nu L_G(d+2)}{2} + \sup_{p\geq 1} \sqrt{\frac{d+p}{p}} \frac{1}{\sqrt{p}} \mathbf{E}_{\xi_i} \left[\|\nabla F(x_t,\xi_i)\|^p\right]^{\frac{1}{p}}$$

$$\leq \frac{\nu L_G(d+2)}{2} + \sup_{p\geq 1} \left(\sqrt{\frac{d+p}{p}} \sup_{p\geq 1} \frac{1}{\sqrt{p}} \mathbf{E}_{\xi_i} \left[\|\nabla F(x_t,\xi_i)\|^p\right]^{\frac{1}{p}}\right)$$

Now,

$$\mathbf{E}_{\xi_i} \left[\|\nabla F(x_t,\xi_i)\|^p\right]^{p^{-1}}$$

$$\leq \mathbf{E}_{\xi_i} \left[(\|\nabla F(x_t,\xi_i) - \nabla f(x_t) + \nabla f(x_t)\|)^p\right]^{p^{-1}}$$

$$\leq \mathbf{E}_{\xi_i} \left[2^{p-1}\|\nabla F(x_t,\xi_i) - \nabla f(x_t)\|^p + 2^{p-1}\|\nabla f(x_t)\|^p\right]^{p^{-1}}$$

$$\leq 2\mathbf{E}_{\xi_i} \left[\|\nabla F(x_t,\xi_i) - \nabla f(x_t)\|^p\right]^{p^{-1}} + 2\|\nabla f(x_t)\|$$

From (5) we have, $\sup_{p\geq 1} p^{-1/2}\mathbf{E}_{\xi_i} \left[(\|\nabla F(x_t,\xi_i) - \nabla f(x_t)\|)^p\right]^{p^{-1}} \leq c_0'\sqrt{\rho-1}\|\nabla_t\|$ where $c_0$ is a constant. Then,

$$\|\phi(\nu,u_i,\xi_i)\|_{\psi_1} \leq \frac{\nu L_G(d+2)}{2} + (2 + c_0'\sqrt{(\rho-1)})\sqrt{d+1}\|\nabla_t\|$$

We also have, $\|u_i^j\|_{\psi_2} \leq 1$, and $\mathbf{E}\left[g_{t,i}\right] = \nabla f_\nu(x_t)$. Then using Lemma A.18, we have $\forall\, j = 1,2,\cdots,d$

$$\mathbb{P}\left(\frac{1}{n_1}\left|\sum_{i=1}^{n_1}\left(g_{t,i}^j - \nabla f_\nu(x_t)^j\right)\right| \geq \tau\right) \leq 4\exp\left(-K_1 \min\left[\left(\frac{\sqrt{n_1}\tau}{\Upsilon_t}\right)^2, \left(\frac{n_1\tau}{\Upsilon_t}\right)^{2/3}\right]\right)$$

where $\Upsilon_t = \frac{\nu L_G(d+2)}{2} + c_0\sqrt{(\rho-1)(d+1)}\|\nabla_t\|$. Using union bound,

$$\mathbb{P}\left(\left\|\frac{1}{n_1}\sum_{i=1}^{n_1}g_{t,i} - \nabla f_\nu(x_t)\right\| \geq \tau\right) \leq \mathbb{P}\left(\exists\, j \in \{1,2,\cdots,d\}s.t.\left|\frac{1}{n_1}\sum_{i=1}^{n_1}g_{t,i}^j - \nabla f_\nu(x_t)^j\right| \geq \tau/\sqrt{d}\right)$$

$$\leq \sum_{j=1}^d \mathbb{P}\left(\left|\frac{1}{n_1}\sum_{i=1}^{n_1}g_{t,i}^j - \nabla f_\nu(x_t)^j\right| \geq \tau/\sqrt{d}\right) \leq 4d\exp\left(-K_1 \min\left[\left(\frac{\sqrt{n_1}\tau}{\Upsilon_t\sqrt{d}}\right)^2, \left(\frac{n_1\tau}{\Upsilon_t\sqrt{d}}\right)^{2/3}\right]\right)$$

Using Lemma A.16 we have

$$\mathbb{P}\left(\left\|\frac{1}{n_1}\sum_{i=1}^{n_1}g_{t,i} - \nabla f(x_t)\right\| \geq \tau\right) \leq \mathbb{P}\left(\left\|\frac{1}{n_1}\sum_{i=1}^{n_1}g_{t,i} - \nabla_\nu f(x_t)\right\| \geq \tau - \frac{\nu L_G(d+3)^{\frac{3}{2}}}{2}\right)$$

∎

**Proof** [Proof of Lemma A.2]

$$\mathbf{E}\left[(c_t\|\zeta_t\|)^{\frac{k}{3}}\right] = \int_0^\infty \mathbb{P}\left((c_t\|\zeta_t\|)^{\frac{k}{3}} > \tau\right)d\tau = \int_0^\infty \mathbb{P}\left(\|\zeta_t\| > \tau^{\frac{3}{k}}/c_t\right)d\tau$$

$$\leq \int_0^\infty 4d\exp(-b_{1,t}\tau'^{2/k})d\tau \leq \int_{-\frac{\nu L_G(d+3)^{\frac{3}{2}}}{2}}^\infty 4d\exp(-b_{1,t}\tau^{2/k})d\tau \leq \int_0^\infty 8d\exp(-b_{1,t}\tau^{2/k})d\tau$$

Substituting, $u = b_{1,t}\tau^{2/k}$ we have,

$$\mathbf{E}\left[(c_t\|\zeta_t\|)^{\frac{k}{3}}\right] \leq \int_0^\infty 4dk b_{1,t}^{-k/2}e^{-u}u^{k/2-1}du = 4dk b_{1,t}^{-k/2}\Gamma\left(k/2\right)$$

Using $2(k!)^2 \leq (2k)!$, and $\Gamma(k+1/2) = (2k)!\sqrt{\pi}/(4^k k!)$, we have

$$\mathbf{E}\left[e^{s(c_t\|\zeta_t\|)^{\frac{1}{3}}}\right] = 1 + \sum_{k=1}^\infty \mathbf{E}\left[\frac{s^k(c_t\|\zeta_t\|)^{\frac{k}{3}}}{k!}\right] \leq 1 + \sum_{k=1}^\infty \frac{s^k}{k!}4dk b_{1,t}^{-k/2}\Gamma\left(k/2\right)$$

$$\leq 1 + 4d\left[\sum_{k=1}^\infty \frac{2ks^{2k}b_{1,t}^{-k}}{(2k)!}\Gamma(k) + \sum_{k=0}^\infty \frac{(2k+1)s^{2k+1}b_{1,t}^{-k-1/2}}{(2k+1)!}\Gamma(k+1/2)\right]$$

$$\leq 1 + 4d \left[ \sum_{k=1}^{\infty} \frac{s^{2k} b_{1,t}^{-k}}{k!} + \sqrt{\frac{\pi s^2}{b_{1,t}}} \sum_{k=0}^{\infty} \frac{s^{2k} b_{1,t}^{-k}}{4^k k!} \right]$$

$$\leq 1 + 4d \left[ e^{\frac{s^2}{b_{1,t}}} + \sqrt{\frac{\pi s^2}{b_{1,t}}} e^{\frac{s^2}{4 b_{1,t}}} \right] \leq 1 + 8d e^{\frac{s^2}{b_{1,t}}} \leq 9d e^{\frac{s^2}{b_{1,t}}}$$

∎

**Proof** [Proof of Lemma A.3] Setting $c = \eta \|\nabla_i\|$, using Lemma A.2 we have

$$\mathbf{E} \left[ e^{s(\eta \|\nabla_i\| \|\zeta_i\|)^{\frac{1}{3}}} \right] \leq 9d e^{\frac{s^2}{b_{1,i}}}.$$

Hence, we have the following:

$$\mathbf{E} \left[ \exp \left( s \sum_{i=0}^{t-1} (\eta \|\nabla_i\| \|\zeta_i\|)^{\frac{1}{3}} - \sum_{i=0}^{t-1} \frac{s^2}{b_{1,i}} \right) \right]$$

$$= \mathbf{E} \left[ \exp \left( s \sum_{i=0}^{t-2} (\eta \|\nabla_i\| \|\zeta_i\|)^{\frac{1}{3}} - \sum_{i=0}^{t-1} \frac{s^2}{b_{1,i}} \right) \mathbf{E} \left[ \exp \left( s(\eta \|\nabla_{t-1}\| \|\zeta_{t-1}\|)^{\frac{1}{3}} \right) | \mathcal{F}_{t-2} \right] \right]$$

$$= 9d \mathbf{E} \left[ \exp \left( s \sum_{i=0}^{t-2} (\eta \|\nabla_i\| \|\zeta_i\|)^{\frac{1}{3}} - \sum_{i=0}^{t-1} \frac{s^2}{b_{1,i}} \right) e^{\frac{s^2}{b_{1,t-1}}} \right]$$

$$= 9d \mathbf{E} \left[ \exp \left( s \sum_{i=0}^{t-2} (\eta \|\nabla_i\| \|\zeta_i\|)^{\frac{1}{3}} - \sum_{i=0}^{t-2} \frac{s^2}{b_{1,i}} \right) \right].$$

Continuing like above we get,

$$\mathbf{E} \left[ \exp \left( s \sum_{i=0}^{t-1} (\eta \|\nabla_i\| \|\zeta_i\|)^{\frac{1}{3}} - \sum_{i=0}^{t-1} \frac{s^2}{b_{1,i}} \right) \right] \leq (9d)^t. \tag{32}$$

Now, we attempt the main result. Note that, we have

$$\mathbb{P} \left( \eta \sum_{i=0}^{t} \nabla_i^{\top} \zeta_i \geq \tau \right) \leq \mathbb{P} \left( \eta \sum_{i=0}^{t} \|\nabla_i\| \|\zeta_i\| \geq \tau \right)$$

$$\leq \mathbb{P} \left( \sum_{i=0}^{t} (\eta \|\nabla_i\| \|\zeta_i\|)^{\frac{1}{3}} \geq \tau^{\frac{1}{3}} \right)$$

$$= \mathbb{P} \left( s \sum_{i=0}^{t} (\eta \|\nabla_i\| \|\zeta_i\|)^{\frac{1}{3}} - \sum_{i=0}^{t-1} \frac{s^2}{b_{1,i}} \geq s \tau^{\frac{1}{3}} - \sum_{i=0}^{t-1} \frac{s^2}{b_{1,i}} \right)$$

$$= \mathbb{P} \left( \exp \left( s \sum_{i=0}^{t} (\eta \|\nabla_i\| \|\zeta_i\|)^{\frac{1}{3}} - \sum_{i=0}^{t-1} \frac{s^2}{b_{1,i}} \right) \geq \exp \left( s \tau^{\frac{1}{3}} - \sum_{i=0}^{t-1} \frac{s^2}{b_{1,i}} \right) \right)$$

$$\leq \frac{\mathbf{E} \left[ \exp \left( s \sum_{i=0}^{t-1} (\eta \|\nabla_i\| \|\zeta_i\|)^{\frac{1}{3}} - \sum_{i=0}^{t-1} \frac{s^2}{b_{1,i}} \right) \right]}{\exp \left( s \tau^{\frac{1}{3}} - \sum_{i=0}^{t-1} \frac{s^2}{b_{1,i}} \right)}$$

$$\leq \exp \left( t \log 9d - s \tau^{\frac{1}{3}} + \sum_{i=0}^{t-1} \frac{s^2}{b_{1,i}} \right).$$

The RHS is minimized at $s = \frac{\tau^{1/3}}{2 \sum_{i=0}^{t-1} \frac{1}{b_{1,i}}}$. Substituting for $s$ this value, for some $l > 0$ we have:

$$t \log 9d - \tau^{\frac{2}{3}} / \left( 4 \sum_{i=0}^{t-1} \frac{1}{b_{1,i}} \right) = -l.$$

Hence, we have

$$\tau = \left( 4 \sum_{i=0}^{t-1} \frac{1}{b_{1,i}} (t \log 9d + l) \right)^{3/2}.$$

Finally, to prove the statement of the Lemma, note that

$$\left( \sum_{i=0}^{t-1} \frac{1}{b_{1,i}} \right)^{\frac{3}{2}} = \frac{\sqrt{d}}{K_1^{\frac{3}{2}} n_1} \left( \sum_{i=0}^{t-1} (c_i \Upsilon_i)^{\frac{2}{3}} \right)^{\frac{3}{2}} \leq \frac{\eta \sqrt{dt}}{K_1^{\frac{3}{2}} n_1} \sum_{i=0}^{t-1} \left( \frac{\nu L_G (d+2)}{2} \|\nabla_i\| + C_0 \sqrt{(\rho-1)(d+1)} \|\nabla_i\|^2 \right)$$

∎

**Proof** [Proof of Lemma A.4] From (32) we have,

$$\mathbf{E} \left[ \exp \left( s \sum_{i=0}^{t-1} (\|\zeta_i\|)^{\frac{1}{3}} - \sum_{i=0}^{t-1} \frac{s^2}{b_{0,i}} \right) \right] \leq (9d)^t$$

where $b_{0,i}$ is as defined in Lemma A.2.

$$\mathbb{P} \left( \sum_{i=0}^{t-1} \|\zeta_i\|^2 \geq \tau \right) \leq \mathbb{P} \left( s \sum_{i=0}^{t-1} \|\zeta_i\|^{\frac{1}{3}} - \sum_{i=0}^{t-1} \frac{s^2}{b_{0,i}} \geq s \tau^{\frac{1}{6}} - \sum_{i=0}^{t-1} \frac{s^2}{b_{0,i}} \right)$$

$$= \mathbb{P} \left( \exp \left( s \sum_{i=0}^{t-1} \|\zeta_i\|^{\frac{1}{3}} - \sum_{i=0}^{t-1} \frac{s^2}{b_{0,i}} \right) \geq \exp \left( s \tau^{\frac{1}{6}} - \sum_{i=0}^{t-1} \frac{s^2}{b_{0,i}} \right) \right)$$

$$\leq \frac{\mathbf{E} \left[ \exp \left( s \sum_{i=0}^{t-1} \|\zeta_i\|^{\frac{1}{3}} - \sum_{i=0}^{t-1} \frac{s^2}{b_{0,i}} \right) \right]}{\exp \left( s \tau^{\frac{1}{6}} - \sum_{i=0}^{t-1} \frac{s^2}{b_{0,i}} \right)} \leq \exp \left( t \log 9d - s \tau^{\frac{1}{6}} + \sum_{i=0}^{t-1} \frac{s^2}{b_{0,i}} \right)$$

Following steps as in Lemma A.3 we have, $\tau = \left( 4 \sum_{i=0}^{t-1} \frac{1}{b_{0,i}} (t \log 9d + l) \right)^3$.

$$\left( \sum_{i=0}^{t-1} \frac{1}{b_{0,i}} \right)^3 = \frac{d}{K_1^3 n_1^2} \left( \sum_{i=0}^{t-1} \Upsilon_i^{\frac{2}{3}} \right)^3 \leq \frac{2dt^2}{K_1^3 n_1^2} \sum_{i=0}^{t-1} \left( \left( \frac{\nu L_G (d+2)}{2} \right)^2 + C_0^2 (\rho-1)(d+1) \|\nabla_i\|^2 \right)$$

∎

**Proof** [Proof of Lemma A.5]

a)

$$f(x_{t+1}) \leq f(x_t) + \nabla_t^\top (x_{t+1} - x_t) + \frac{L_G}{2} \|x_{t+1} - x_t\|^2$$

$$\leq f(x_t) - \eta \nabla_t^\top (\nabla_t + \tilde{\zeta}_t) + \frac{\eta^2 L_G}{2} \left( \frac{3}{2} \|\nabla_t\|^2 + 3 \|\tilde{\zeta}_t\|^2 \right)$$

$$\leq f(x_t) - \frac{\eta}{4} \|\nabla_t\|^2 - \eta \nabla_t^\top \tilde{\zeta}_t + \frac{3\eta^2 L_G}{2} \|\tilde{\zeta}_t\|^2$$

The last inequality holds as we will choose $\eta \leq 1/L_G$. Summing both sides,

$$f(x_t) - f(x_0) \leq -\frac{\eta}{4} \sum_{i=0}^{t-1} \|\nabla_t\|^2 - \eta \sum_{i=0}^{t-1} \nabla_i^\top \tilde{\zeta}_i + \frac{3\eta^2 L_G}{2} \sum_{i=0}^{t-1} \|\tilde{\zeta}_i\|^2 \qquad (33)$$

Observe that, by Assumption 2.4,

$$\mathbb{P}(\nabla_t^\top \zeta_t \geq \tau | \mathcal{F}_{t-1}) \leq \mathbb{P}(\|\nabla_t\| \|\zeta_t\| \geq \tau | \mathcal{F}_{t-1}) \leq 2 \exp(-\tau^2 / (\frac{2(\rho-1)}{n_1} \|\nabla_t\|^4)) \quad (34)$$

So $\nabla_t^\top \zeta_t | \mathcal{F}_{t-1}$ is $c \sqrt{\frac{\rho-1}{n_1}} \|\nabla_t\|^2$-subGaussian. Using Lemma A.13, we have, with probability at least $1 - e^{-l}$,

$$-\eta \sum_{i=0}^{t-1} \nabla_i^\top \zeta_i \leq \lambda \eta c \frac{\rho-1}{n_1} \sum_{i=0}^{t-1} \|\nabla_i\|^4 + \eta \frac{l}{\lambda} \leq \lambda \eta c \frac{\rho-1}{n_1} \left( \sum_{i=0}^{t-1} \|\nabla_i\|^2 \right)^2 + \eta \frac{l}{\lambda}$$

Plugging $\lambda = \frac{32l}{\sum_{i=0}^{t-1}\|\nabla_i\|^2}$, we have,

$$-\eta\sum_{i=0}^{t-1}\nabla_i^\top\zeta_i \le \eta\left(32cl\frac{\rho-1}{n_1}+\frac{1}{32}\right)\sum_{i=0}^{t-1}\|\nabla_i\|^2 \tag{35}$$

Using Lemma A.13, with probability at least $1-e^{-l}$ we have,

$$-\eta\sum_{i=0}^{t-1}\nabla_i^\top\theta_i \le \frac{\eta}{32}\sum_{i=0}^{t-1}\|\nabla_i\|^2 + 32cl\eta r^2 \tag{36}$$

Using Lemma A.14, we have with probability at least $1-e^{-l}$,

$$\sum_{i=0}^{t-1}\|\theta_i\|^2 \le cr^2(t+l) \tag{37}$$

Note that by Assumption 2.4, $\mathbf{E}\left[\|\zeta_t\|^2|\mathcal{F}_{t-1}\right] \le \frac{\rho-1}{n_1}\|\nabla_t\|^2$, and $\|\zeta_t\|^2|\mathcal{F}_{t-1}$ is $c\frac{\rho-1}{n_1}\|\nabla_t\|^2$-subExponential. So we have, with probability at least $1-e^{-l}$,

$$\sum_{i=0}^{t-1}\|\zeta_i\|^2 \le (c+l)\frac{\rho-1}{n_1}\sum_{i=0}^{t-1}\|\nabla_i\|^2 \tag{38}$$

Combining (33), (35), (36), (37) and (38), using $\|\tilde{\zeta}_t\|^2 \le 2(\|\zeta_t\|^2+\|\theta\|^2)$, and using union bound, we have with probability at least $1-4e^{-l}$,

$$f(x_t) - f(x_0) \le \left(-\frac{\eta}{4}+\eta\left(32lc\frac{\rho-1}{n_1}+\frac{1}{32}\right)+\frac{\eta}{32}+3\eta^2 L_G(c+l)\frac{\rho-1}{n_1}\right)\sum_{i=0}^{t-1}\|\nabla_i\|^2$$
$$+ 3c\eta^2 r^2(t+l)L_G + 32cl\eta r^2$$

We need to choose $\eta$ such that $\left(-\frac{\eta}{4}+\eta\left(32lc\frac{\rho-1}{n_1}+\frac{1}{32}\right)+\frac{\eta}{32}+3\eta^2 L_G(c+l)\frac{\rho-1}{n_1}\right) < -\frac{\eta}{16}$. Choosing $n_1$, and $\eta$ as in (18), and setting $t = \mathcal{T}_1$, we get (19).

b) Using Lemma A.3, and Lemma A.4, and Assumption 2.1 we have, with probability at least $1-4e^{-l}$

$$f(x_t) - f(x_0) \le -\frac{\eta}{4}\sum_{i=0}^{t-1}\|\nabla_i\|^2 + \frac{\eta}{16}\sum_{i=0}^{t-1}\|\nabla_i\|^2 + 16cl\eta r^2 + 3cL_G\eta^2 r^2(t+l)$$

$$+\frac{8\eta\sqrt{dt}}{K_1^{\frac{3}{2}}n_1}(t\log 9d+l)^{\frac{3}{2}}\sum_{i=0}^{t-1}\left(\frac{\nu LL_G(d+2)}{2}+C_0\sqrt{(\rho-1)(d+1)}\|\nabla_i\|^2\right)$$

$$+\frac{384L_G d\eta^2 t^2(t\log 9d+l)^3}{K_1^3 n_1^2}\sum_{i=0}^{t-1}\left(\left(\frac{\nu L_G(d+2)}{2}\right)^2+C_0^2(\rho-1)(d+1)\|\nabla_i\|^2\right)$$

We will choose $\mathcal{T}_0$, $\eta$, and $n_1$ such that, (20), and (21) are true. Then, with probability at least $1-4e^{-l}$, we get (22).

∎

**Proof** [Proof of Lemma A.6]

a) For a fixed $\tau \le t$, we have

$$\|x_\tau - x_0\|^2 \le \eta^2\|\sum_{i=0}^{\tau-1}(\nabla_i+\tilde{\zeta}_i)\|^2 \le 2\eta^2 t\sum_{i=0}^{t-1}\|\nabla_i\|^2 + 4\eta^2(\|\sum_{i=0}^{t-1}\zeta_i\|^2+\|\sum_{i=0}^{t-1}\theta_i\|^2)$$

Using Lemma A.15, we have with probability at least $1-4de^{-l}$,

$$\|\sum_{i=0}^{t-1}\zeta_i\|^2 + \|\sum_{i=0}^{t-1}\theta_i\|^2 \le cl\left(\frac{\rho-1}{n_1}\sum_{i=0}^{t-1}\|\nabla_i\|^2 + tr^2\right)$$

Combining this with Lemma A.5, with probability at least $1-4e^{-l}-4de^{-l}$, setting $t = \mathcal{T}_1$, and using union bound we have (23).

b)

$$\mathbb{P}\left(\|\sum_{i=0}^{t-1}\zeta_i\|^2 \geq \tau\right) \leq \mathbb{P}\left(\sum_{i=0}^{t-1}\|\zeta_i\|^2 \geq \tau/t\right)$$

So from Lemma A.4, we have with probability at least $1 - e^{-l}$

$$\|\sum_{i=0}^{t-1}\zeta_i\|^2 \leq \frac{128dt^3(t\log 9d + l)^3}{K_1^3 n_1^2}\sum_{i=0}^{t-1}\left(\left(\frac{\nu L_G(d+2)}{2}\right)^2 + C_0^2(\rho-1)(d+1)\|\nabla_i\|^2\right)$$

Plugging $t = \mathcal{T}_0$, under condition (20), we have,

$$\|\sum_{i=0}^{\mathcal{T}_0-1}\zeta_i\|^2 \leq \frac{L_G\mathcal{T}_0^2\nu^2(d+2)^2}{192C_0^2(\rho-1)(d+1)} + \frac{\mathcal{T}_0}{12L_G}\sum_{i=0}^{\mathcal{T}_0-1}\|\nabla_i\|^2$$

From (22) we have, with probability at least $1 - e^{-l}$

$$\sum_{i=0}^{\mathcal{T}_0-1}\|\nabla_i\|^2 \leq \frac{16}{\eta}(f(x_0) - f(x_{\mathcal{T}_0}) + \wp(r, l, \nu, \eta, d, \mathcal{T}_0))$$

Then we have with probability with at least $1 - 3d\mathcal{T}_0 e^{-l}$, we have (24).

■

**Proof** [Proof of Lemma A.10]

a) Proof for the first-order setting is as in [25].

b) Note that $q_h(t)$ is the same as in part (a). If we can ensure that for the zeroth-order case $\forall t \leq \mathcal{T}_0$ we have $\|q_{sg}(t+1)\| \leq \beta(t)r/(40\sqrt{d})$, then the rest of the proof follows from [25]. For a fixed $t$, using Cauchy–Schwarz inequality,

$$\mathbb{P}\left(\|q_{sg}(t+1)\| \geq \tau\right) = \mathbb{P}\left(\eta\left\|\sum_{i=0}^{t}(I - \eta\mathcal{H})^{t-i}\hat{\zeta}_i\right\| \geq \tau\right) \leq \mathbb{P}\left(\eta\sum_{i=0}^{t}\|(I - \eta\mathcal{H})\|^{t-i}\|\zeta_i - \zeta_i'\| \geq \tau\right)$$

$$\leq 2\mathbb{P}\left(\eta\sum_{i=0}^{t}\|(I - \eta\mathcal{H})\|^{t-i}\|\zeta_i\| \geq \tau/2\right)$$

$$\leq 2\mathbb{P}\left(\eta\sqrt{\sum_{i=0}^{t}\|(I - \eta\mathcal{H})\|^{2t-2i}}\sqrt{\sum_{i=0}^{t}\|\zeta_i\|^2} \geq \tau/2\right)$$

$$\leq 2\mathbb{P}\left(\sum_{i=0}^{t}\|\zeta_i\|^2 \geq \left(\frac{\tau}{2\eta\beta(t+1)}\right)^2\right)$$

From Lemma A.4, we have with probability at least $1 - e^{-l}$

$$\|q_{sg}(t+1)\| \leq \frac{32\sqrt{d(t+1)}\eta\beta(t+1)((t+1)\log 9d + \log 2 + l)^{\frac{3}{2}}}{K_1^{3/2}n_1}$$

$$\sum_{i=0}^{t}\left(\frac{\nu L_G(d+2)}{2} + C_0\sqrt{(\rho-1)(d+1)}\|\nabla_i\|\right)$$

Recalling the definition of $\Xi$ from (30), and setting $t = \mathcal{T}_0$, we will choose $\mathcal{T}_0$, $r$, $\eta$, $l$, and $\nu$ such that

$$\Xi \cdot \sum_{i=0}^{\mathcal{T}_0}\left(\frac{\nu L_G(d+2)}{2} + C_0\sqrt{(\rho-1)(d+1)}\|\nabla_i\|\right) \leq \frac{\beta(\mathcal{T}_0)r}{40\sqrt{d}} \qquad (39)$$

■

**Proof** [Proof of Lemma A.12]

a) For the first part, we have from Lemma A.5, with probability at least $1 - 4e^{-l}$,

$$f(x_{\mathcal{T}_1}) - f(x_0) \leq 3c\eta^2 r^2(\mathcal{T}_1 + l)L_G + 32cl\eta r^2 \leq 0.1\mathcal{F}_1$$

By similar methods in [25], we have, with probability at least $2/3 - 10d\mathcal{T}_1{}^2 \log\left(\frac{\mathcal{S}_1\sqrt{d}}{\eta r}\right) e^{-l}$, if $\min\{f(x_{\mathcal{T}_1}) - f(x_0), f(x'_{\mathcal{T}_1}) - f(x_0)\} \geq -\mathcal{F}_1$, then

$$\max\{\|x_{\mathcal{T}_1} - x_0\|, \|x'_{\mathcal{T}_1} - x_0\|\} \geq \frac{\beta(\mathcal{T}_1)\eta r}{40\sqrt{d}} = \frac{(1 + \eta\gamma)^{\mathcal{T}_1}\sqrt{\eta}r}{40\sqrt{2\gamma d}} > \mathcal{S}_1 \qquad (40)$$

This is in contradiction with Lemma A.11. Then we have with probability at least $2/3 - 10d\mathcal{T}_1{}^2 \log\left(\frac{\mathcal{S}_1\sqrt{d}}{\eta r}\right) e^{-l}$, $\min\{f(x_{\mathcal{T}_1}) - f(x_0), f(x'_{\mathcal{T}_1}) - f(x_0)\} \leq -\mathcal{F}_1$. As the marginal distributions of $x_{\mathcal{T}_1}$ and $x'_{\mathcal{T}_1}$ are same we have,

$$\mathbb{P}(f(x'_{\mathcal{T}_1}) - f(x_0)) \leq -\mathcal{F}_1) \geq \frac{1}{2}\mathbb{P}(\min\{f(x_{\mathcal{T}_1}) - f(x_0), f(x'_{\mathcal{T}_1}) - f(x_0)\} \leq -\mathcal{F}_1)$$

$$\geq 1/3 - 9d\mathcal{T}_1{}^2 \log\left(\frac{\mathcal{S}_1\sqrt{d}}{\eta r}\right) e^{-l}$$

b) Note that the probability for the second statement being true is at least $1/3 - 1.5\mathcal{T}_0{}^2 e^{-l}$ which is different from [25] but the proof method is same. So we omit the proof here.

■

# B Proof of Theorem 4.1

We first state the following optimality conditions for CR Newton method updates due to [37].

**Lemma B.1** *[37]*

$$g_t + H_t h_t^* + \frac{M}{2}\|h_t^*\|h_t = 0 \qquad (41a)$$

$$H_t + \frac{M}{2}\|h_t^*\|I \succcurlyeq 0 \qquad (41b)$$

Intuitively, the proof follows through three stages. First, in Lemma B.2, we show that the descent at each time point is proportional to the cube of the step size.

**Lemma B.2** *[47] Let $m_t$ be as defined in* (14). *Then for all t,*

$$m_t(x_t + h_t^*) - m_t(x_t) \leq -\frac{M}{12}\|h_t^*\|^3 \qquad (42)$$

Then, in Lemma B.3 we show that the second-order staionarity of an iterate is upper bounded by the step size at that time point.

**Lemma B.3** *Let Assumption 2.2, and 2.3 hold true for $f$. Then the following holds $\forall t$*

a) *for the first-order update of a CR Newton method,*

$$\sqrt{\mathbf{E}\left[\|h_t^*\|^2|\mathcal{F}_t\right]} \geq \max\left(\left(A\mathbf{E}\left[\|\nabla f\left(x_t + h_t^*\right)\||\mathcal{F}_t\right] - B\right)^{\frac{1}{2}},\right.$$

$$\left.\frac{2}{M + 2L_H}\left(-\sqrt{\frac{\sigma_2^2}{n_2}} - \mathbf{E}\left[\lambda_{1,t+1}|\mathcal{F}_t\right]\right)\right) \qquad (43)$$

*where $A = \frac{1}{2(L_H+M)}\left(1 - \sqrt{\frac{\rho-1}{n_1}}\right)$, and $B = \frac{1}{4(L_H+M)^2}\left(\frac{\rho-1}{2n_1}L_G^2 + \frac{\sigma_2^2}{n_2}\right)$.*

b) *for the zeroth-order update of a CR Newton Method*

$$\sqrt{\mathbf{E}\left[\|h_t^*\|^2|\mathcal{F}_t\right]} \geq \max\left((A'\mathbf{E}\left[\|\nabla f\left(x_t + h_t^*\right)\||\mathcal{F}_t\right] - B')^{\frac{1}{2}}, \right.$$

$$\left. \frac{2}{M + 2L_H}\left(-\sqrt{\frac{128(1 + 2\log 2d)(d + 16)^4 L_G^2}{3n_2}} - \sqrt{3}\nu L_H(d + 16)^{\frac{5}{2}} - \mathbf{E}\left[\lambda_{1,t+1}|\mathcal{F}_t\right]\right)\right)$$

(44)

*where $A' = \frac{1}{2(L_H + M)}\left(1 - \sqrt{\frac{\rho' - 1}{n_1}}\right)$, and*
$B' = \frac{1}{4(L_H + M)^2}\left(\frac{\rho' - 1}{n_1}L_G^2 + \frac{128(1 + 2\log 2d)(d + 16)^4 L_G^2}{3n_2} + 3L_H^2\nu^2(d + 16)^5 + \sqrt{6}\nu(L_H + M)L_G(d + 3)^{\frac{3}{2}}\right).$

Finally, in Lemma B.4, we prove that the expected step size becomes smaller with the horizon.

**Lemma B.4** *Let $f$ be a function for which Assumptions 2.2, and 2.3 are true. Then,*

a) *for first-order updates generated by Algorithm 2 the following holds:*

$$\left(\frac{M}{72} - \left(\frac{\rho - 1}{n_1}\right)^{\frac{3}{4}}\frac{8}{\sqrt{M}A^{\frac{3}{2}}}\right)\mathbf{E}\left[\|h_R^*\|^3|\mathcal{F}_t\right]$$

$$\leq \frac{f(x_1) - f^*}{T} + \frac{1152L_G^3}{M^2}\left(\frac{\rho - 1}{n_1}\right)^{\frac{3}{2}}$$

$$+ \frac{8}{\sqrt{M}}\left(\frac{\rho - 1}{n_1}\right)^{\frac{3}{4}}\left(\frac{B}{A}\right)^{\frac{3}{2}} + \frac{324}{M^2}\frac{\sigma_2^3}{n_2^{3/2}}$$

(45)

*where $R$ is an integer random variable uniformly distributed over the support $\{1, 2, \cdots, T\}$.*

b) *for zeroth-order updates generated by Algorithm 2 the following holds:*

$$\left(\frac{M}{144} - \left(\frac{\rho' - 1}{n_1}\right)^{\frac{3}{4}}\frac{6}{\sqrt{M}A'^{\frac{3}{2}}}\right)\mathbf{E}\left[\|h_R^*\|^3|\mathcal{F}_t\right]$$

$$\leq \frac{f(x_1) - f^*}{T} + \frac{864L_G^3}{M^2}\left(\frac{\rho' - 1}{n_1}\right)^{\frac{3}{2}} + \frac{4}{M}(\nu L_G)^{\frac{3}{2}}(d + 3)^{\frac{9}{4}}$$

$$+ \frac{6}{\sqrt{M}}\left(\frac{\rho' - 1}{n_1}\right)^{\frac{3}{4}}\left(\frac{B'}{A'}\right)^{\frac{3}{2}} + \frac{162}{M^2}\left(\frac{160\sqrt{1 + 2\log 2d}(d + 16)^6 L_G^3}{n_2^{\frac{3}{2}}} + 21L_H^3(d + 16)^{\frac{15}{2}}\nu^3\right)$$

(46)

*where $R$ is an integer random variable uniformly distributed over the support $\{1, 2, \cdots, T\}$.*

Combining the above three facts, we complete proof of Theorem 4.1.

**Proof** [Proof of Theorem 4.1]

a) From Lemma B.3 we have,

$$\sqrt{\mathbf{E}\left[\|h_t^*\|^2|\mathcal{F}_t\right]} + \sqrt{B} + \frac{2}{(2L_H + M)}\sqrt{\frac{\sigma_2^2}{n_2}} \geq$$

$$\max\left(\sqrt{A\mathbf{E}\left[\|\nabla f\left(x_t + h_t^*\right)\||\mathcal{F}_t\right]}, -\frac{2}{(2L_H + M)}\mathbf{E}\left[\lambda_{1,t+1}|\mathcal{F}_t\right]\right)$$

(47)

From Lemma B.4, we have

$$\left(\left(\frac{M}{72} - \left(\frac{\rho - 1}{n_1}\right)^{\frac{3}{4}}\frac{8}{\sqrt{M}A^{\frac{3}{2}}}\right)\mathbf{E}\left[\|h_R^*\|^3|\mathcal{F}_t\right]\right)^{\frac{1}{3}}$$

$$\leq \left(\frac{f(x_1) - f^*}{T}\right)^{\frac{1}{3}} + \frac{11L_G}{M^{\frac{2}{3}}}\left(\frac{\rho - 1}{n_1}\right)^{\frac{1}{2}}$$

$$+ \frac{2}{M^{\frac{1}{6}}}\left(\frac{\rho - 1}{n_1}\right)^{\frac{1}{4}}\left(\frac{B}{A}\right)^{\frac{1}{2}} + \frac{7}{M^{\frac{2}{3}}}\frac{\sigma_2}{n_2^{1/2}} \tag{48}$$

Combining (47) with (48), using Jensens's inequality we have, and choosing $n_1$, $n_2$, $T$, and $M$ as in (15), we have $\max\left(\sqrt{\frac{\mathbf{E}[\|\nabla f(x_R)\|]}{144M}}, -\frac{\mathbf{E}[\lambda_{1,R}]}{9M}\right) \leq \sqrt{\epsilon}$. Total number of first-order oracle calls, and second-order oracle calls are $Tn_1 = Tn_2 = \mathcal{O}\left(\frac{1}{\epsilon^{\frac{5}{2}}}\right)$.

b) From Lemma B.3 we have,

$$\sqrt{\mathbf{E}\left[\|h_t^*\|^2|\mathcal{F}_t\right]} + \sqrt{B'} + \frac{2}{(2L_H + M)}\left(\sqrt{\frac{128(1 + 2\log 2d)(d + 16)^4 L_G^2}{3n_2}} + \sqrt{3}\nu L_H(d + 16)^{\frac{5}{2}}\right) \geq$$

$$\max\left(\sqrt{A'\mathbf{E}\left[\|\nabla f(x_t + h_t^*)\||\mathcal{F}_t\right]}, -\frac{2}{(2L_H + M)}\mathbf{E}\left[\lambda_{1,t+1}|\mathcal{F}_t\right]\right) \tag{49}$$

From Lemma B.4, we have

$$\left(\left(\frac{1}{144} - \left(\frac{\rho' - 1}{n_1}\right)^{\frac{3}{4}}\frac{6}{M^{\frac{3}{2}}A'^{\frac{3}{2}}}\right)\mathbf{E}\left[\|h_R^*\|^3|\mathcal{F}_t\right]\right)^{\frac{1}{3}}$$

$$\leq \left(\frac{f(x_1) - f^*}{MT}\right)^{\frac{1}{3}} + \frac{10L_G}{M}\left(\frac{\rho' - 1}{n_1}\right)^{\frac{1}{2}} + \frac{2}{M^{\frac{2}{3}}}(\nu L_G)^{\frac{1}{2}}(d + 3)^{\frac{3}{4}}$$

$$+ \frac{2}{M^{\frac{1}{2}}}\left(\frac{\rho' - 1}{n_1}\right)^{\frac{1}{4}}\left(\frac{B'}{A'}\right)^{\frac{1}{2}} + \frac{6}{M}\left(\frac{6L_G(1 + 2\log 2d)^{\frac{1}{6}}(d + 16)^2}{n_2^{\frac{1}{2}}} + 3L_H(d + 16)^{\frac{5}{2}}\nu\right) \tag{50}$$

Combining (49) with (50), using Jensens's inequality we have, and choosing $n_1$, $n_2$, $T$, $\nu$, and $M$ as in (16), we have $\max\left(\sqrt{\mathbf{E}\left[\|\nabla f(x_R)\|\right]}, -\mathbf{E}\left[\lambda_{1,R}\right]\right) \leq \mathcal{O}\left(\sqrt{\epsilon}\right)$. Total number of first-order oracle calls is $Tn_1 = \mathcal{O}\left(\frac{d}{\epsilon^{\frac{5}{2}}}\right)$, and second-order oracle calls is $Tn_2 = \mathcal{O}\left(\frac{d^4\log d}{\epsilon^{\frac{5}{2}}}\right)$.

∎

## B.1 Proofs of Lemmas related to CR Newton method

**Lemma B.5** *[42]*

$$\mathbf{E}\left[\left\|\nabla_t^2 - \frac{1}{n_2}\sum_{i=1}^{n_2}\nabla^2 F(x_t, \xi_i)\right\|^2\right] \leq \frac{\sigma_2^2}{n_2} \tag{51}$$

$$\mathbf{E}\left[\left\|\nabla_t^2 - \frac{1}{n_2}\sum_{i=1}^{n_2}\nabla^2 F(x_t, \xi_i)\right\|^3\right] \leq \frac{2\sigma_2^3}{n_2^{\frac{3}{2}}} \tag{52}$$

For the zeroth-order estimates of gradient and Hessian as defined in (3) we have the following concentration result.

**Lemma B.6** *[5]*

$$\mathbf{E}\left[\|g_t - \nabla_t\|^2\right] \leq \frac{\rho' - 1}{n_1}\|\nabla_t\|^2 + \frac{3\nu^2}{2}L_G^2(d + 3)^3 \tag{53a}$$

$$\mathbf{E}\left[\|\nabla_t^2 - H_t\|^2\right] \le \frac{128(1 + 2\log 2d)(d+16)^4 L_G^2}{3n_2} + 3L_H^2(d+16)^5 \nu^2 \tag{53b}$$

$$\mathbf{E}\left[\|\nabla_t^2 - H_t\|^3\right] \le \frac{160\sqrt{1 + 2\log 2d}(d+16)^6 L_G^3}{n_2^{\frac{3}{2}}} + 21L_H^3(d+16)^{\frac{15}{2}} \nu^3 \tag{53c}$$

*where* $\rho' = 1 + 4(d+5)\rho$

**Proof** [Proof of Lemma B.3]

a) Using (41a) we get,

$$\|g_t + H_t h_t^*\| = \frac{M}{2}\|h_t^*\|^2$$

Then, using Assumption 2.3, and Young's inequality we get,

$$\|\nabla f\left(x_t + h_t^*\right)\|$$
$$\le \|\nabla f\left(x_t + h_t^*\right) - \nabla_t - \nabla_t^2 h_t^*\| + \|\nabla_t + \nabla_t^2 h_t^*\|$$
$$\le \|\nabla f\left(x_t + h_t^*\right) - \nabla_t - \nabla_t^2 h_t^*\| + \|g_t + H_t h_t^*\|$$
$$+ \|g_t - \nabla_t\| + \|\left(H_t - \nabla_t^2\right) h_t^*\|$$
$$\le \frac{M + L_H}{2}\|h_t^*\|^2 + \|g_t - \nabla_t\| + \|\left(H_t - \nabla_t^2\right) h_t^*\|$$
$$\le (M + L_H)\|h_t^*\|^2 + \|g_t - \nabla_t\| + \frac{1}{2(L_H + M)}\|H_t - \nabla_t^2\|^2$$

Taking expectation on both sides, and using Lemma 2.1, Lemma B.5, and Jensen's inequality we have

$$\mathbf{E}\left[\|\nabla f\left(x_t + h_t^*\right)\| | \mathcal{F}_t\right]$$
$$\le (L_H + M)\mathbf{E}\left[\|h_t^*\|^2 | \mathcal{F}_t\right] + \sqrt{\frac{\rho - 1}{n_1}}\|\nabla_t\| + \frac{\sigma_2^2}{2(L_H + M)n_2}$$

Using Assumption 2.2 we get

$$\left(1 - \sqrt{\frac{\rho - 1}{n_1}}\right)\mathbf{E}\left[\|\nabla f\left(x_t + h_t^*\right)\| | \mathcal{F}_t\right]$$
$$\le (L_H + M)\mathbf{E}\left[\|h_t^*\|^2 | \mathcal{F}_t\right] + \sqrt{\frac{\rho - 1}{n_1}}L_G \mathbf{E}\left[\|h_t^*\| | \mathcal{F}_t\right] + \frac{\sigma_2^2}{2(L_H + M)n_2}$$
$$\le 2(L_H + M)\mathbf{E}\left[\|h_t^*\|^2 | \mathcal{F}_t\right] + \frac{1}{2(L_H + M)}\left(\frac{\rho - 1}{n_1}L_G^2 + \frac{\sigma_2^2}{n_2}\right)$$

Rearranging we have,

$$\sqrt{\mathbf{E}\left[\|h_t^*\|^2 | \mathcal{F}_t\right]}$$
$$\ge \left(\frac{1}{2(L_H + M)}\left(1 - \sqrt{\frac{\rho - 1}{n_1}}\right)\mathbf{E}\left[\|\nabla f\left(x_t + h_t^*\right)\| | \mathcal{F}_t\right]\right.$$
$$\left. - \frac{1}{4(L_H + M)^2}\left(\frac{\rho - 1}{n_1}L_G^2 + \frac{\sigma_2^2}{n_2}\right)\right)^{\frac{1}{2}} \tag{54}$$

Now, using Assumption 2.3 we get

$$\mathbf{E}\left[\nabla^2 f\left(x_t + h_t^*\right) | \mathcal{F}_t\right] \succcurlyeq \mathbf{E}\left[\nabla_t^2 - L_H \|h_t^*\| I | \mathcal{F}_t\right]$$
$$\succcurlyeq \mathbf{E}\left[H_t - L_H \|h_t^*\| I | \mathcal{F}_t\right] - \sqrt{\frac{\sigma_2^2}{n_2}}I$$
$$\succcurlyeq -\sqrt{\frac{\sigma_2^2}{n_2}}I - \left(L_H + \frac{M}{2}\right)\mathbf{E}\left[\|h_t^*\| | \mathcal{F}_t\right]I$$

$$\mathbf{E}\left[\|h_t^*\| \mid \mathcal{F}_t\right] \geq \frac{2}{M + 2L_H}\left(-\sqrt{\frac{\sigma_2^2}{n_2}} - \mathbf{E}\left[\lambda_{1,t+1} \mid \mathcal{F}_t\right]\right) \tag{55}$$

Now using Jensen's inequality, and (54) we get (43).

b) Using Lemma B.6, and following the proof of part (a), (54) becomes

$$\sqrt{\mathbf{E}\left[\|h_t^*\|^2 \mid \mathcal{F}_t\right]} \geq \left(\frac{1}{2(L_H + M)}\left(1 - \sqrt{\frac{\rho' - 1}{n_1}}\right)\mathbf{E}\left[\|\nabla f\left(x_t + h_t^*\right)\| \mid \mathcal{F}_t\right]\right.$$

$$-\frac{1}{4(L_H + M)^2}\left(\frac{\rho' - 1}{n_1}L_G^2 + \frac{128(1 + 2\log 2d)(d + 16)^4 L_G^2}{3n_2} + 3L_H^2\nu^2(d + 16)^5 + \sqrt{6}\nu(L_H + M)L_G(d + 3)^{\frac{3}{2}}\right)$$

$$\tag{56}$$

Similarly, (55) becomes

$$\mathbf{E}\left[\|h_t^*\| \mid \mathcal{F}_t\right] \geq \frac{2}{(2L_H + M)}\left(-\sqrt{\frac{128(1 + 2\log 2d)(d + 16)^4 L_G^2}{3n_2}} - \sqrt{3}\nu L_H(d + 16)^{\frac{5}{2}} - \mathbf{E}\left[\lambda_{1,t+1} \mid \mathcal{F}_t\right]\right)$$

$$\tag{57}$$

∎

**Proof** [Proof of Lemma B.4]

a) Using Young's inequality, and (42), we get

$$f(x_t + h_t^*) - f(x_t) \leq m_t(x_t + h_t^*) - m_t(x_t)$$
$$+ (\nabla_t - g_t)^\top h_t^* + \frac{1}{2}h_t^{*\top}(\nabla_t^2 - H_t)h_t^*$$
$$\leq m_t(x_t + h_t^*) - m_t(x_t)$$
$$+ \frac{4}{\sqrt{3M}}\|\nabla_t - g_t\|^{\frac{3}{2}} + \frac{162}{M^2}\|\nabla_t^2 - H_t\|^3 + \frac{M}{18}\|h_t^*\|^3$$
$$\leq -\frac{M}{36}\|h_t^*\|^3 + \frac{4}{\sqrt{3M}}\|\nabla_t - g_t\|^{\frac{3}{2}} + \frac{162}{M^2}\|\nabla_t^2 - H_t\|^3$$

Taking expectation on both sides, and using Lemma 2.1 with Jensen's inequality, and Lemma B.5, we get

$$\mathbf{E}\left[f(x_t + h_t^*) \mid \mathcal{F}_t\right] - f(x_t) \leq -\frac{M}{36}\mathbf{E}\left[\|h_t^*\|^3 \mid \mathcal{F}_t\right]$$
$$+ \frac{4}{\sqrt{3M}}\left(\frac{\rho - 1}{n_1}\right)^{\frac{3}{4}}\|\nabla_t\|^{\frac{3}{2}} + \frac{162}{M^2}\frac{2\sigma_2^3}{n_2^{3/2}} \tag{58}$$

Now let us relate the gradient size $\|\nabla_t\|$ with $\|h_t^*\|$. Note that, as $x_{t+1} = x_t + h_t^*$ we will use $\nabla_{t+1}$ to denote $\nabla f(x_t + h_t^*)$ here. Using triangle inequality, the fact $(a + b)^{3/2} \leq \sqrt{2}(a^{3/2} + b^{3/2})$ for $a, b > 0$, Assumption 2.2, and Jensen's inequality we get

$$\|\nabla_t\|^{\frac{3}{2}} = \|\nabla_t - \mathbf{E}\left[\nabla_{t+1} \mid \mathcal{F}_t\right] + \mathbf{E}\left[\nabla_{t+1} \mid \mathcal{F}_t\right]\|^{\frac{3}{2}}$$
$$\leq(\|\nabla_t - \mathbf{E}\left[\nabla_{t+1} \mid \mathcal{F}_t\right]\| + \|\mathbf{E}\left[\nabla_{t+1} \mid \mathcal{F}_t\right]\|)^{\frac{3}{2}}$$
$$\leq\sqrt{2}(\|\nabla_t - \mathbf{E}\left[\nabla_{t+1} \mid \mathcal{F}_t\right]\|^{\frac{3}{2}} + \|\mathbf{E}\left[\nabla_{t+1} \mid \mathcal{F}_t\right]\|^{\frac{3}{2}})$$
$$\leq\sqrt{2}(L_G^{\frac{3}{2}}\mathbf{E}\left[\|h_t^*\|^{\frac{3}{2}} \mid \mathcal{F}_t\right] + \mathbf{E}\left[\|\nabla_{t+1}\| \mid \mathcal{F}_t\right]^{\frac{3}{2}}) \tag{59}$$

From Lemma B.3 we have,

$$\mathbf{E}\left[\|h_t^*\|^2 \mid \mathcal{F}_t\right] + B \geq A\mathbf{E}\left[\|\nabla_{t+1}\| \mid \mathcal{F}_t\right]$$

Again using the fact $(a + b)^{3/2} \leq \sqrt{2}(a^{3/2} + b^{3/2})$ for $a, b > 0$, and Jensens's inequality we get

$$\sqrt{2}\left(\mathbf{E}\left[\|h_t^*\|^3 \mid \mathcal{F}_t\right] + B^{\frac{3}{2}}\right) \geq \left(A\mathbf{E}\left[\|\nabla_{t+1}\| \mid \mathcal{F}_t\right]\right)^{\frac{3}{2}} \tag{60}$$

Combining (59), and (60), we get

$$\|\nabla_t\|^{\frac{3}{2}} \leq \sqrt{2} L_G^{\frac{3}{2}} \mathbf{E}\left[\|h_t^*\|^{\frac{3}{2}}|\mathcal{F}_t\right] + \frac{2}{A^{\frac{3}{2}}} \mathbf{E}\left[\|h_t^*\|^3|\mathcal{F}_t\right]$$

$$+ 2\left(\frac{B}{A}\right)^{\frac{3}{2}}$$

Now, using Young's inequality

$$\left(\frac{\rho-1}{n_1}\right)^{\frac{3}{4}} \|\nabla_t\|^{\frac{3}{2}} \leq \frac{288 L_G^3}{M^{\frac{3}{2}}} \left(\frac{\rho-1}{n_1}\right)^{\frac{3}{2}}$$

$$+ \frac{\sqrt{3} M^{\frac{3}{2}}}{288} \mathbf{E}\left[\|h_t^*\|^3|\mathcal{F}_t\right] + \left(\frac{\rho-1}{n_1}\right)^{\frac{3}{4}} \frac{2}{A^{\frac{3}{2}}} \mathbf{E}\left[\|h_t^*\|^3|\mathcal{F}_t\right]$$

$$+ 2\left(\frac{\rho-1}{n_1}\right)^{\frac{3}{4}} \left(\frac{B}{A}\right)^{\frac{3}{2}} \tag{61}$$

Combining (58), and (61) we get

$$\mathbf{E}\left[f(x_t + h_t^*)|\mathcal{F}_t\right] - f(x_t) \leq -\frac{M}{72} \mathbf{E}\left[\|h_t^*\|^3|\mathcal{F}_t\right]$$

$$+ \frac{1152 L_G^3}{M^2} \left(\frac{\rho-1}{n_1}\right)^{\frac{3}{2}}$$

$$+ \left(\frac{\rho-1}{n_1}\right)^{\frac{3}{4}} \frac{8}{\sqrt{M} A^{\frac{3}{2}}} \mathbf{E}\left[\|h_t^*\|^3|\mathcal{F}_t\right]$$

$$+ \frac{8}{\sqrt{M}} \left(\frac{\rho-1}{n_1}\right)^{\frac{3}{4}} \left(\frac{B}{A}\right)^{\frac{3}{2}} + \frac{324}{M^2} \frac{\sigma_2^3}{n_2^{3/2}} \tag{62}$$

Rearranging and summing from $t = 1$ to $T$, and dividing both sides by $T$ we get (45).

b) Using Lemma B.6, and following the proof of Lemma B.4 we have the following inequality corresponding to (58)

$$\mathbf{E}\left[f(x_t + h_t^*)|\mathcal{F}_t\right] - f(x_t) \leq -\frac{M}{36} \mathbf{E}\left[\|h_t^*\|^3|\mathcal{F}_t\right]$$

$$+ \frac{3}{\sqrt{M}} \left(\frac{\rho'-1}{n_1}\right)^{\frac{3}{4}} \|\nabla_t\|^{\frac{3}{2}} + \frac{4}{M} (\nu L_G)^{\frac{3}{2}} (d+3)^{\frac{9}{4}}$$

$$+ \frac{162}{M^2} \left(\frac{160\sqrt{1 + 2\log 2d}(d+16)^6 L_G^3}{n_2^{\frac{3}{2}}} + 21 L_H^3 (d+16)^{\frac{15}{2}} \nu^3\right) \tag{63}$$

Eventually we get the following descent in the function value similar to (62)

$$\mathbf{E}\left[f(x_t + h_t^*)|\mathcal{F}_t\right] - f(x_t) \leq -\frac{M}{144} \mathbf{E}\left[\|h_t^*\|^3|\mathcal{F}_t\right]$$

$$+ \frac{864 L_G^3}{M^2} \left(\frac{\rho'-1}{n_1}\right)^{\frac{3}{2}} + \left(\frac{\rho'-1}{n_1}\right)^{\frac{3}{4}} \frac{6}{\sqrt{M} A'^{\frac{3}{2}}} \mathbf{E}\left[\|h_t^*\|^3|\mathcal{F}_t\right] + \frac{4}{M} (\nu L_G)^{\frac{3}{2}} (d+3)^{\frac{9}{4}}$$

$$+ \frac{6}{\sqrt{M}} \left(\frac{\rho'-1}{n_1}\right)^{\frac{3}{4}} \left(\frac{B'}{A'}\right)^{\frac{3}{2}} + \frac{162}{M^2} \left(\frac{160\sqrt{1 + 2\log 2d}(d+16)^6 L_G^3}{n_2^{\frac{3}{2}}} + 21 L_H^3 (d+16)^{\frac{15}{2}} \nu^3\right) \tag{64}$$

Rearranging and summing from $t = 1$ to $T$, and dividing both sides by $T$ we get (46).

■