[Reviews · NeurIPS 2020]

Review 1

Summary and Contributions: This work studied the convergence of two popular methods: SGD and cubic regularization, under a special condition called strong-growth condition (SGC). The authors suggested that for nonconvex optimization, SGC helps SGD and cubic regularization find the approximate local minimizer faster. The authors also showed that such an acceleration also holds for the case where only zeroth-order oracle is accessible.

Strengths: - The organization of this work is clear. - The theoretical results in this paper are sound.

Weaknesses: - The importance of the SGC condition remains unclear. In Line 129, the authors claimed that SGC condition is satisfied in some practical settings such as the training of deep neural networks, therefore the SGC condition should be regarded as an interesting special setting for nonconvex optimization. However, recent work [1,2] showed that the training of deep neural networks can be further regarded as a special task of convex optimization in the Neural tangent kernel (NTK) regime, which is a stronger condition than SGC. Therefore, the authors may want to clarify the importance of SGC by showing some more examples in machine learning. - The technique contribution of this work seems incremental. As the authors suggested, [VBS18] firstly studied the SGC condition under nonconvex setting and proposed that SGD costs O(1/\epsilon^2) gradient complexity to find first-order stationary points. Meanwhile, note that [AZL18] proposed a generic framework which could turn any algorithms for finding first-order stationary points into algorithms for finding approximate local minimizer, without hurting the convergence rate. Therefore, is it true that the convergence rate O(1/\epsilon^2) for SGD to find approximate local minimizer established in this paper can be directly deduced by combining existing results from [VBS18] and [AZL18]? The authors may want to discuss above issues to highlight their technique contribution in this paper. - The authors may want to add some discussion about how the SGC condition will affect the variance-reduction based algorithms such as [3,4,5,6]. [1] Allen-Zhu, Zeyuan, Yuanzhi Li, and Zhao Song. "A convergence theory for deep learning via over-parameterization." International Conference on Machine Learning. 2019. [2] Zou, Difan, et al. "Gradient descent optimizes over-parameterized deep ReLU networks." Machine Learning 109.3 (2020): 467-492. [3] Johnson, Rie, and Tong Zhang. "Accelerating stochastic gradient descent using predictive variance reduction." Advances in neural information processing systems. 2013. [4] Zhou, Dongruo, Pan Xu, and Quanquan Gu. "Stochastic nested variance reduction for nonconvex optimization." Advances in Neural Information Processing Systems. 2018. [5] Wang, Zhe, et al. "SpiderBoost and momentum: Faster variance reduction algorithms." Advances in Neural Information Processing Systems. 2019. [6] Nguyen, Lam M., et al. "Finite-sum smooth optimization with sarah." arXiv preprint arXiv:1901.07648 (2019).

Correctness: The claims and methods proposed by the authors are correct.

Clarity: This paper is written well and easy to follow.

Relation to Prior Work: This work should discuss more about previous work to show the difference.

Reproducibility: No

Additional Feedback: ########################### I am OK with the authors' response.


Review 2

Summary and Contributions: This work analyzes the oracle complexity of perturbed stochastic gradient descent algorithm and the stochastic cubic-regularized Newton’s method for escaping saddle-points in nonconvex stochastic optimization. They show that under interpolation-like conditions these two algorithms obtain improved rates for escaping saddle-points.

Strengths: The authors proved inproved results of speed of PSGD and SCRN under new assumptions.

Weaknesses: The extra assumption is not explained very well and lacks of intuitions. As the core assumption, the authors just explain a little about this assumption.

Correctness: I think the claims are correct.

Clarity: Yes.

Relation to Prior Work: Clear.

Reproducibility: Yes

Additional Feedback: This work analyzes the oracle complexity of perturbed stochastic gradient descent algorithm and the stochastic cubic-regularized Newton’s method for escaping saddle-points in nonconvex stochastic optimization. They show that under interpolation-like conditions these two algorithms obtain improved rates for escaping saddle-points. The proofs are based on the Assumption 3.1. 1. More details of Assumption 3.1 needs to be presented. The previous SGD just used the unbiased asssumtion and bounded variance. This paper uses a detailed distribution of the stochastic gradient, thus, proves the better results. Can this assumption be satisfied in the network training or finite sum optimization. 2. The results and proofs are quite complicated. The authors may present the proof sketch to make this paper more readable for the readers. 3. As a theoretical paper, the authors may present the core techniques of the proofs. 4. The zeroth and first order methods are meaningful and common in the ML. The high order methd needs the Hessian matrix and cannot be applied to high dimensional case. I think it is better to remove the results of high order methd into the supplementary part. ------------------------------------------------------------------------------------------------- comments after rebuttal ------------------------------------------------------------------------------------------------- I have read your feedback. Thank you!


Review 3

Summary and Contributions: This work improves the convergence rate of PSGD and SCRN by using strong growth condition. Further, the theoretical analysis is extended to the zeroth-order variants of PSGD and SCRN.

Strengths: (1) It is great to see that the convergence rate of PSGD and SCRN is improved by using strong growth condition. (2) The theoretical analysis is established for both zeroth-order and high-order methods.

Weaknesses: (1) The theoretical contributions in this paper are not significantly strong. Most of the theoretical analysis follows directly from previous works, e.g., [JNG+19], [VBS18], by assuming the strong growth condition. Taking that into account, the technical contributions of this work are not significant. (2) The strong growth condition pushes all the stochastic samples of the total loss function to share the same critical point, which makes the stochastic analysis much simpler due to the automatic variance reduction. Therefore, the strong growth condition is actually a very strong assumption. Although the authors mention that the per-sample loss in the supervised learning seems to satisfy the strong growth condition, but looking carefully, this requires there is a ground-truth model/classifier to exactly fit the training data. What if there is label noise in the training? In other words, if the strong growth condition is approximately satisfied, then do we still achieve such improved convergence rate? More discussions are encouraged in terms of reasonability and generality of the strong growth condition. (3) The numerical experiments are weak. No numerical experiments are developed to verify the proposed upper-bounds of the sample-complexity/oracle calls. I am caring about the experiments because many assumptions used in this work that are hard to verify in practice, more discussions/experiments are needed to clarify what practical scenarios fall into the framework of this paper.

Correctness: Seems correct. No empirical results are provided.

Clarity: Yes.

Relation to Prior Work: Yes.

Reproducibility: Yes

Additional Feedback: =================After response======================= I read the author response, and agree with the authors for the responses for my part.

[Author Response · NeurIPS 2020]

We thank all the reviewers for their valuable comments. We first highlight our technical contributions.

**Main technical contributions:** We emphasize that our main contribution is the first result showing that PSGD and
SCRN escape saddle-points and converge to local minimizers faster under Strong Growth Condition (SGC) (which
is a consequence of interpolation phenomenon). Prior works (e.g., [VBS18]) considered only convergence to critical
points under SGC in the first-order setting. We provide our results in both the zeroth and higher order settings. In
the zeroth-order setting, we prove a novel concentration inequality for the zeroth-order gradient estimator which is
non-trivial and was not know before. We emphasize that this concentration result does not assume the function is
bounded; see also Remark 3. For completeness, we also analyzed the complexity for zeroth-order PSGD without the
SGC assumption for unbounded functions, which was not done before in the literature. Furthermore, the analysis of
SCRN is also significantly involved under SGC (especially in zeroth-order setup); see also Remark 6 and 7.

**Rev #1: NTK regime:** NTK viewpoint provides an alternative explanation on efficiency of optimizing algorithms
for DNN training. However, a majority of the results based on NTK approach are for polynomially (in depth and
sample-size) large-width networks ([1] mention that their polynomial degrees are impractical). Landscape analysis of
DNN training (which, roughly speaking is: all local min are global min and so escaping saddle-points and converging
to local-min is needed) provides an alternative view for finite-width multilayer neural networks (see e.g., **P2** and its
references). Our contributions in this paper are geared towards the later angle – as DNNs are also interpolators in
practice, we show that under SGC we can converge faster to local-min. **Other examples:** Another concrete example
satisfying SGC condition is online matrix completion (see **P1** for details). We will add this example in detail in our
revision. **Contributions:** Please see Lines 2-10 above. **[AZL18]:** At a high-level, the suggested approach would also
work. However, the method in [AL18] is a theoretical computer science style reduction approach. It involves a wrapper
algorithm on top of PSGD which increases overall runtime. Our result directly analyzes PSGD iterates, more in line
with optimization and machine learning type results. Furthermore, a main drawback of the approach in [AZL18] is that
it is not directly applicable for the 0th-order setting due to their bounded variance (of stochastic gradient) assumption.
Please also see the discussion in Remark 5 for more details. **Variance reduction (VR):** A motivation for the SGC
assumption is that it provides *automatic VR*. As in Table 1, for stochastic setting under SGC, PSGD already achieves the
corresponding complexity of its deterministic counterpart (without acceleration). It is interesting future work to examine
if similar results hold for finite-sum setting. However, most VR methods invariably involve double-loop algorithms and
their empirical performance in deep learning has come under close scrutiny in the recent past; see [DB19] and [Sch20].

**Rev #2: Intuitions:** SGC at a high-level could intuitively be described as an assumption satisfied in interpolation
models, providing *automatic variance reduction*. Please also see Lemma 3.1 and Remark 1 for more intuition. We
will clarify this more in our revision. **Detailed distribution of stochastic grad:** We clarify that *we do not make any*
*distributional assumption* on the stochastic gradient. SGC is only a variance/moment based assumption. **Finite-sum**
**opt:** Finite-sum opt is a special case of stochastic setting and hence the assumptions are satisfied; see also [MBB18,
VBS18]. **Proof Sketch:** Thanks for this suggestion. We will add a proof sketch in our revision. **High-order method:**
Making higher-order methods practical is an interesting future work. We will clarify this point and reorganize.

**Rev #6: Theoretical contributions:** Please see Lines 2-10 above. **Approximate SGC:** Thanks for raising this
extremely interesting question. Considering 1st-order setting, departures from SGC can be modeled, for example, by:
$E(\|\nabla F(x_t, \xi)\|^2) \leq \rho \|\nabla f(x_t)\|^2 + e_t^2$ where $e_t^2 > 0$ is an additive non-vanishing iteration-dependent noise variance;
see also [VBS18]. This assumption could be used for example to model certain specific types of label-noise. When
$e_t^2 = \sigma^2, \forall t$, the oracle complexity is $\tilde{O}(\epsilon^{-4})$ since this case is essentially equivalent to the standard stochastic gradient
setting and SGC has no effect. But one could obtain rates approaching $\tilde{O}(\epsilon^{-2})$ depending on the decay of the term
$R_t = \sum_{i=1}^t e_i^2$. If we assume $R_t \approx t^\alpha$ for $\alpha \in (-\infty, 1]$, then we can show that the complexity is $\tilde{O}(\epsilon^{-\max(2, 3.5+0.5\alpha)})$.
Hence, when $\alpha = -2$, the complexity is $\tilde{O}(\epsilon^{-2.5})$ and when $\alpha \leq -3$, it is $\tilde{O}(\epsilon^{-2})$. In this setup, it may be possible to
connect label noise (for specific noise models) to $\alpha$. This would provide a possible way of incorporating label noise
in this setup. It is intriguing to examine this problem rigorously as future work. We will clarify this in our revision.
**Experiments:** We will be happy to add simulations and real-world experiments on DNN and Online matrix completion
in our revision. We clarify that our main goal in this work is to provide a plausible explanation for the question *why*
*do optimization algorithms for deep-learning models work efficiently in practice despite the associated non-convexity*
*?*. As an attempt, [MBB18], [VBS18] and related works proposed the SGC condition which is satisfied due to the
interpolating nature of DNNs. However, prior works fell short of providing a complete explanation as they only analyze
convergence to critical points. Recently, it has been shown in several works that for DNNs (in the finite-width regime),
all approximate local minima are also global minima (see reference **P2**) due to which converging to local-min and
escaping saddle-points are important. Hence, based on these motivations, we show that PSGD and SCRN converge to
local-min faster with SGC condition. We emphasize that this line of work is only one plausible explanation for the
above question and there are other directions (e.g., NTK) attempting alternative explanations in the literature.

**References: P1** - *Provable Efficient Online Matrix Completion via Non-convex Stochastic Gradient Descent*, Jin et al.,
NeurIPS 2016. **P2** - Elimination of All Bad Local Minima in Deep Learning, Kawaguchi et al., AISTATS, 2020.

[Meta-Review · NeurIPS 2020]

This paper gives faster convergence guarantees for non-convex stochastic optimization under an assumption called strong growth condition. The reviewers found the theoretical analysis to be novel and interesting. The paper could be improved if it can provide further evidence that SGC is satisfied for objective functions of interest (in particular whether SGC will be satisfied near saddle points and whether that's necessary for the guarantee are unclear).